



# Ice volume and basal topography estimation using geostatistical methods and ground-penetrating radar measurements: application to the Tsanfleuron and Scex Rouge glaciers, Swiss Alps

**Alexis Neven**[1,★]**, Valentin Dall'Alba**[1,★]**, Przemysław Juda**[1]**, Julien Straubhaar**[1]**, and Philippe Renard**[1,2]

[1]Centre of Hydrogeology and Geothermics, University of Neuchâtel, Neuchâtel, Switzerland
[2]Department of Geosciences, University of Oslo, Oslo, Norway
★These authors contributed equally to this work.

**Correspondence:** Alexis Neven (alexis.neven@unine.ch)

**Abstract.** Ground-penetrating radar (GPR) is widely used for determining mountain glacier thickness. However, this method provides thickness data only along the acquisition lines, and therefore interpolation has to be made between them. Depending on the interpolation strategy, calculated ice volumes can differ and can lack an accurate error estimation. Furthermore, glacial basal topography is often characterized by complex geomorphological features, which can be hard to reproduce using classical interpolation methods, especially when the field data are sparse or when the morphological features are too complex. This study investigates the applicability of multiple-point statistics (MPS) simulations to interpolate glacier bedrock topography using GPR measurements. In 2018, a dense GPR data set was acquired on the Tsanfleuron Glacier (Switzerland). These data were used as the source for a bedrock interpolation. The results obtained with the direct-sampling MPS method are compared against those obtained with kriging and sequential Gaussian simulations (SGSs) on both a synthetic data set – with known reference volume and bedrock topography – and the real data underlying the Tsanfleuron Glacier. Using the MPS modeled bedrock, the ice volume for the Scex Rouge and Tsanfleuron glaciers is estimated to be $113.9 \pm 1.6$ million cubic meters. The direct-sampling approach, unlike the SGS and kriging, allowed not only an accurate volume estimation but also the generation of a set of realistic bedrock simulations. The complex karstic geomorphological features are reproduced and can be used to significantly improve for example the precision of subglacial flow estimation.

## 1 Introduction

It is widely accepted that global climatic changes impact future precipitation rates and temperatures. In Switzerland, these changes will inevitably induce new stresses on alpine environments and on glacier mass balance (e.g., Haeberli et al., 2007; Beniston, 2012; Huss and Fischer, 2016). In this context, the monitoring of glaciers' thickness and volume is crucial in order to predict their melt rate and the possible consequences for water resources, sediment production, and slope stabilization.

Ice volume estimation relies on two components: (1) the surface topography of the glacier and (2) the underlying bedrock topography. The first one is easily measurable using lidar (e.g., Haugerud et al., 2003), satellite measures (e.g., Berthier et al., 2014), or uncrewed aerial vehicle (UAV) (e.g., Chudley et al., 2019). However, the variations in basal topography are difficult to measure, due to the impossibility of reaching it easily with direct measurements. Ground-penetrating radar (GPR) is widely used for determining the thickness of ice masses (Flowers and Clarke, 1999; Plewes and Hubbard, 2001; Bohleber et al., 2017). The equipment has the advantage of being light and easy to use in a glacial environment. However, this method only provides thickness data along the acquisition lines, and therefore interpolation methods are needed to estimate the basal topography between sparse survey lines. Depending on the interpolation methods, the basal topography can change significantly and can lead to a wide range of calculated ice volumes (e.g.,

Gabbi et al., 2012). Moreover, some of the methods generally used are unable to provide an accurate error estimation. Furthermore, if we are interested in the basal topography to simulate subglacial water flow, for example, the choice of the method becomes critical since the subglacial water flow process is highly non-linearly linked to the morphology of the subglacial topography.

One classical interpolation strategy used for basal estimation is the ordinary kriging method (e.g., Vanlooy et al., 2014). This method provides fast and reliable interpolation of the data and returns the best linear unbiased estimator. The kriging estimation produces a smooth interpolation and does not represent the possible detailed morphology of the bedrock when it is not constrained by sufficient data. Furthermore, even if kriging allows estimation of the local point-wise standard deviation of the simulated value, here the elevation of the bedrock, it cannot be used to estimate the uncertainty of the global volume of ice. The standard deviation of a non-linear process, estimated from a kriged variable, cannot be simply computed from the kriging standard deviation map (see, e.g., Chiles and Delfiner, 2012, p. 478). Stochastic simulations using Gaussian processes (SGS) have the opposite aim to represent the variability and spatial statistics of the simulated variable of interest, here the basal topography (Goff et al., 2014). The simulations can be used for uncertainty estimation. A downside of these two methods is that they are based on two-point spatial statistics, usually a variogram model. The variogram quantifies the spatial continuity of the data. It shows how well two points at a given distance from each other are correlated. Different types of variogram models can be fitted to the experimental data. These methods rely on the use of a multi-Gaussian random function model. This assumption implies then that the simulated fields belong to a given class of models and cannot simulate all possible complex spatial patterns when the data are not sufficiently dense.

Other empirical methods using volume–area ($V–A$) relations are also used to calculate ice volume. These methods include slope-dependent volume estimation, ice thickness distribution (e.g., Frey et al., 2014), or surface velocity estimation (e.g., Gantayat et al., 2014). $V–A$ estimation methods are based on the observation that larger glaciers tend to be thicker than smaller ones. These methods are easy and fast to apply. However, they generally lack spatial uncertainty analysis and can be very sensitive to their parameterization processes. Their application results in a single ice volume estimation, which does not allow the uncertainty in the model to be captured. Finally, they only produce an estimation of the thickness of the glacier and cannot help to predict subglacial topography.

In the last decades, new geostatistical methods have arisen with the aim to improve the realism of the simulation using another form of information than the one expressed by two-point statistics and variogram interpretation. Multiple-point statistics (MPS) simulation algorithms use a training image

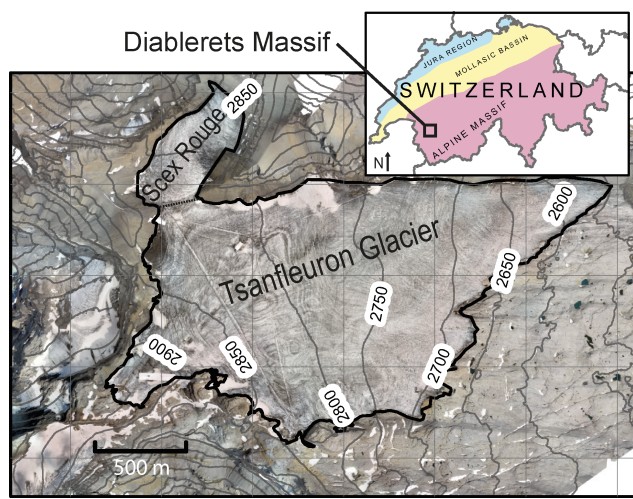

**Figure 1.** Aerial image and digital elevation model captured from drone images of the Tsanfleuron Glacier, Switzerland.

(TI) to infer the spatial statistics of the model and generate random fields reproducing these spatial statistics. The TI represents a conceptual knowledge of the variable that is aimed to be simulated. It can be created by experts or can be extracted from analog data sets. Unlike other geostatistical approaches, MPS does not require the definition of an analytical two-point statistics model to represent the spatial variability but instead infers it implicitly from the TI provided by the user (Journel and Zhang, 2006). MPS simulations benefit from this additional knowledge and can also be constrained by the acquired data (the conditioning data). These methods allow the creation of more realistic spatial patterns than classical two-point geostatistical methods and can be used to represent the uncertainty by simulating a set of realizations. Some examples of application can be found in Oriani et al. (2014), de Carvalho et al. (2016), and Dall'Alba et al. (2020). In recent studies, MPS has successfully been applied to estimate subglacial topography from synthetic data (MacKie and Schroeder, 2020) and to evaluate the probability of subglacial lakes (MacKie et al., 2020a).

The aims of this paper are both methodological and applied. Regarding the methodological aspect, this work aims to demonstrate the use of the MPS method to combine information provided by GPR data points and a digital elevation model (DEM) to simulate a realistic glacial basal topography. The benefits of using a MPS approach are highlighted by comparing its results with those obtained with more classical geostatistical methods. Using synthetic test cases, the different methods are compared by calculating for each one an ice volume and a roughness estimation, which are then compared against the true synthetic values. A set of scores are computed to compare the methods. Through this comparison process different parameter sets are also tested for each method. This methodological aspect helps to select the most suitable parameter sets on a synthetic case where the tar-

get topography is known. This highlights the advantages of the multiple-point simulation approach for estimating glacier volume and its associated basal geomorphology. On the applied side, the objectives are to present new field data and new estimations of the volume of the Tsanfleuron Glacier, roughly 10 years after the last detailed published estimation (Gremaud and Goldscheider, 2010).

The Tsanfleuron Glacier and the Scex Rouge Glacier (Fig. 1) are both located in the Diablerets Massif (Schoeneich and Reynard, 2021) in Switzerland. They are connected to each other at the Tsanfleuron pass, but are located on two different faces: the main slope of the Scex Rouge Glacier is toward the north-northeast, while the Tsanfleuron slope is toward the east. Tsanfleuron volume was estimated to be 100 million cubic meters of ice in 2009 using radio-magneto-telluric data and kriging of the ice thickness, with an uncertainty of $\pm 10\%$ and a maximum measured depth of 138 m (Gremaud and Goldscheider, 2010). According to the measurements, the glacier is currently thinning by $\sim 1.5\,\mathrm{m\,yr^{-1}}$ (Gremaud and Goldscheider, 2010). A more recent publication, applied to all Swiss glaciers, proposed a volume for Tsanfleuron and Scex Rouge of respectively 200.02 and 8.12 million cubic meters in 2016 (Grab et al., 2021). However, the uncertainty on the GPR picking used in the particular case of Tsanfleuron is important, especially with thicknesses bigger than 60 m according to personal communication with one of the authors. Both glaciers lie on carbonate formations that are heavily karstified. The glacier is one of the main feeders of the underlying karstic system. Tracer tests showed that this network is a significant source of drinking water supply for the community of Conthey (Gremaud, 2008). Obtaining a better model of the basal topography is therefore a step toward improving the understanding of the remaining ice volume, the glacial retreat behavior, and its impact on the regional groundwater system.

The core MPS technique used in this study is the direct-sampling algorithm (Mariethoz et al., 2010) implemented in the *DeeSse* code (Straubhaar, 2019). This implementation includes several improvements compared to the original algorithm. In particular DeeSse can account for multi-resolution structures in the data set (Straubhaar et al., 2020) and inequality data (Straubhaar and Renard, 2021).

In this paper, the exposed basal surface of the melted glacier zones is employed as a training image for the simulation of the covered glacier basal topography. The justification for this is that the lithology and general topographical slope below the glacier and in the exposed area are similar, and therefore the geomorphological features should also be similar. This idea is validated by the analysis of the area where the glacier has retreated in the last dozen years, which exposed geomorphological structures similar to the older part. The GPR-inferred depths are then used as conditioning points as well as the topographical data around the glacier.

Since the exact topography below the glacier is unknown, to analyze and benchmark the performances of different in-

terpolation methods, a numerical experiment was designed in which references can be compared to the simulation outputs. For that purpose, the exposed part of the bedrock is also used (besides being used as TI) to compare the performances of the MPS, kriging, and sequential Gaussian simulation (SGS) approaches. In total 20 zones are extracted from the exposed DEM and sampled to create fake GPR data sets. Using only the sampled data set, the topography in the test zones are interpolated using MPS, SGS, and kriging and compared to the reference topography. The true volumes of the synthetic tests are defined as being the space between the simulated topography and a flat surface representing the top of the ice sheet. The altitude of this ice sheet is defined as being 4 m above the maximum altitude of the simulation. The absolute volumes of the simulations are then compared to this true volume, using the same ice sheet altitude. Moreover, an estimation of hydrological flow accumulation is calculated on the simulated bedrock and on the reference. The flow accumulation map outlines the link between structures of the topography and the connectivity of the cells. In addition, different parameter sets are tested for each method through this experiment. Finally, different scores are used to compare the methods. This numerical experiment helps to understand and visualize the impact of each method on the simulated bedrock shape and volume estimation distribution.

Lastly, the Tsanfleuron and Scex Rouge bedrock's topography is interpolated using the previously tested methods and parameter sets. The simulated topography is also compared to recently uncovered bedrock, using a simple estimation of flow accumulation. A brief overview of the glacier volume distributions and their past evolution is finally carried out using the calculated bedrock surfaces and a different DEM.

## 2 Methods

### 2.1 GPR and DEM acquisition

In summer 2018, a dense GPR acquisition on the Tsanfleuron Glacier was performed (Fig. 2) using a single Radarteam Cobra GPR antenna of 80 Mhz center frequency, mounted on a backpack with a real-time kinematic (RTK) differential GPS. Total listening time was set to 1600 ns with a sample rate of 320 MHz. The GPR data were processed using a standard workflow. A time-zeroing was performed, setting the origin of the time vector when the first arrival is recorded. Our system being a single-antenna system, this time corresponds in fact to the recording of the pulse itself. We then apply a time-dependent gain, the time vector raised to a power of 1.2. A de-wow filter was also added, using the residual median filter method described in Gerlitz et al. (2008). We then removed the mean trace to avoid displaying the pulse and the airwave present in all the traces. Finally, a 120 MHz low-pass filter was applied to remove the signals outside of the GPR band. The data were binned in a 2 m grid. The time-

to-depth conversion was done using a uniform wave propagation speed of $0.168\,\mathrm{m\,ns^{-1}}$ (Eisen et al., 2002; Moorman and Michel, 2000). The basal reflection identification and the picking were then performed multiple times, with a random display of the lines, by four different operators. The random selection of the line was done in order to avoid bias and over-interpretation of the GPR data reflections. The processing was identical for all operators; however they had the possibility to adapt the display (color map and percentile clipping). Any of the picked points which showed differences of more than 5 m between the different operators were considered unsure and therefore rejected. This resulted in a good basal depth estimation in 87 % of the lines. As expected, the deepest zones (more than 70–80 m) are the most difficult to identify precisely, with a smaller signal-to-noise ratio. An example line is displayed in Fig. 2. The data were then converted to a point set, representing the position of the measurement and the altitude of the basal reflector, to be used with the MPS algorithm.

In addition, during the summer in 2019 several UAV flights were undertaken above the Tsanfleuron Glacier. We conducted five flights with a Sense Fly EBEE UAV equipped with a 20 megapixel RGB camera. The objective was to have a resolution of at least $10\,\mathrm{cm\,px^{-1}}$ everywhere on the glacier and an overlap of 60 % between the images. The resulting flight altitude was between 300 and 600 m above ground. A DEM and an ortho-mosaic (Fig. 1) were generated using a stereoscopic method, and the whole domain was geolocated using ground control points. The image processing was done using the commercial Pix4D software.

## 2.2 Multiple-point simulation

The MPS algorithm used in this paper is DeeSse (Straubhaar, 2019). It is based on the direct-sampling technique that is described in detail in Mariethoz et al. (2010). The principle of the method is that a conditional simulation is generated by sampling patterns from a training data set. The simulation is sequential; a random path is used to define in which order the cells of the grid have to be simulated. For each cell, the pattern constituted by the already simulated or conditioning data surrounding the current cell is retrieved. The algorithm then searches in the training data and, in a random manner, some patterns that are similar to the conditioning pattern. When a similar pattern is found, the value at the central location of the pattern is copied from the training data to the simulation grid. This technique allows us to jointly co-simulate several variables, meaning that secondary information can be used to improve the simulations. The code proposes a set of options such as the relative distance search (Mariethoz et al., 2010), allowing us to account for non-stationarity of the mean values in the simulation grid. Furthermore, the DeeSse code includes several improvements compared to the original method such as the use of Gaussian pyramids (Straubhaar et al., 2020). This feature simulates patterns at different

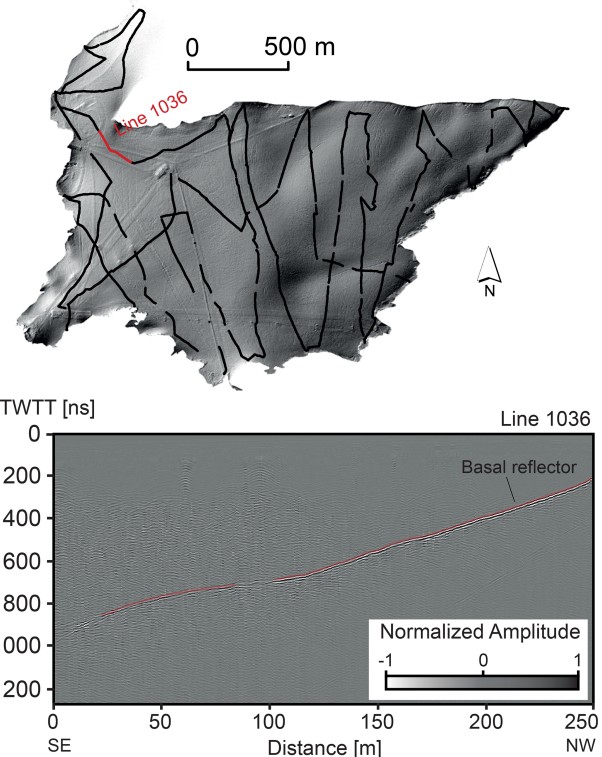

**Figure 2.** Acquisition lines from August 2018. The cumulative length is about 18 km. Hill shade from the DEM derived from drone images. The straight lines visible on the hill shade are ski lifts and transit tracks for snow groomers. A GPR line is displayed, with the basal reflector outlined.

scales, to ensure that both large- and small-scale patterns are well reproduced in the simulation.

The three main parameters of the method are the maximum number of neighbors ($n$), the distance threshold ($t$), and the scan fraction ($f$). $n$ controls how many nodes are used for the pattern comparison between the TI and the simulation grid. $t$ sets the maximum acceptable dissimilarity when comparing a pattern in the simulation grid and in the TI. Finally, $f$ controls the maximal fraction of the TI that can be scanned when searching for a pattern. The optimal values of these parameters depend on the complexity of the patterns that are displayed in the training data set and on the acceptable computing time to obtain a set of simulations. Some recommendations on how to select those parameters are presented in Meerschman et al. (2013).

In the case of the Tsanfleuron Glacier, the acquired DEM of the exposed bedrock is used as the TI. The conditioning data are the GPR lines and the altitude of the bedrock (from the DEM) surrounding the glacier. To obtain the best parameters for this data set, a series of experiments with different parameter sets were conducted. The methodology to conduct these tests is detailed in Sect. 2.4.

For all the simulations, the multi-scale mode using Gaussian pyramids and relative distance options is activated to get the best reproduction of the patterns. The first feature improves the simulation of patterns at different scales and produces more consistent simulation outputs (Straubhaar et al., 2020). The second feature is a way to deal with non-stationary data sets. Indeed, we are interested here in the relative changes of the topography along altitude. Two patterns that show the same relative changes even at different absolute altitudes should be considered similar. This option has the advantage of being able to deal with non-stationary TI, without any prior de-trending needed. Finally, a secondary variable is used. This variable is not the main variable of interest, but it is simulated jointly to improve the quality of the simulation. Once some points are simulated, the secondary variable guides the simulation of the neighboring points, reducing the uncertainty and improving the realism. In our case, the topography gradient is added as a secondary variable in the TI. This variable is not defined in the hard data set.

## 2.3 Kriging and SGS

Kriging and sequential Gaussian simulations (SGSs) are standard geostatistical techniques that are well described in many textbooks (e.g., Chiles and Delfiner, 2012). Therefore, we will not describe the underlying theory of these methods here but rather focus on the specific aspects of their application in our case.

At the scale of the study site, the exposed bedrock topography presents a general slope toward the east and is therefore non-stationary. To account for this general slope, we decided to first remove the trend present in the DEM and in the hard (GPR) data before conducting the variogram analysis. To do so, a polynomial surface is fitted and removed from the data. By removing the polynomial surface from the data, only the deviation from this surface, which corresponds to the deviation from the general slope of the glacier, is simulated. At the end of the process, the trend is then added to the interpolated values to obtain the final basal topography. It is important to note that since the Scex Rouge and the Tsanfleuron glaciers have two different orientations, two different trends were interpolated and applied to each glacier, the transition being set at Tsanfleuron pass. The polynomial interpolated trends in the form of

$$f(x, y) = a + bx + cy + dx^2 + exy + fy^2, \tag{1}$$

$$= 0 - 0.281x - 0.143y - 6.09 \times 10^{-6}x^2 + 7.98$$
$$\times 10^{-5}xy - 1.28 \times 10^{-5}y^2 \text{ (Tsanfleuron),} \tag{2}$$

$$= 0 + 1.78x - 3.37y + 4.63 \times 10^{-3}x^2 - 3.26 \times 10^{-3}xy$$
$$+ 1.505 \times 10^{-3}y^2 \text{ (ScexRouge),}$$
$$\tag{3}$$

with $a, b, c, d, e$, and $f$ being the coefficients and $x$ and $y$ the coordinates in the plane. The variogram model used for

the SGS and kriging approaches is shown in Fig. 3. The data set used for calculating the experimental variogram is shown in Fig. 3a in a map view. It is composed of 7500 points, 2500 coming from the acquired GPR data and the other 5000 points randomly sampled from the TI. This data set presents a distribution centered around $-4.08$ m with a standard deviation value of $21.2$ m (Fig. 3b) that is close to Gaussian. We therefore did not apply any variable transform to ensure Gaussianity. The variogram map (Fig. 3c) does not show a strong and significant anisotropy. Therefore, an omnidirectional variogram model is selected and adjusted to the experimental variogram data. The model is composed of two components: a spherical structure with a sill value of $344$ m$^2$, a range value of $862$ m, and a second exponential structure defined by a sill value of $134$ m$^2$ and a range value of $396$ m (Fig. 3d). The sequential Gaussian simulations and kriging are performed using the SGeMS open-source software (Remy et al., 2009).

## 2.4 Systematic comparison of the methods

In order to benchmark the different geostatistical algorithms and parameter sets, a systematic testing approach is applied (Fig. 4). First, the available DEM (the exposed part of the bedrock) is sampled to create 20 synthetic test cases. These are $800 \times 800$ m$^2$ wide zones that are randomly selected in the DEM (Fig. 4). Once the zones are selected, two synthetic GPR acquisition lines are randomly extracted from the topography of each zone. The bed elevation is sampled along the line, and random noise is added to the data. The purpose of this step is to simulate the uncertainty associated with the picking and the processing. The points are then used as conditioning data. The rest of the data are removed. The three interpolation methods are then tested to infer the basal topography. For each zone and for each parameter set, 40 MPS simulations, 40 SGS simulations, and 1 ordinary kriging estimate are performed.

For the MPS approach, nine sets of parameters are considered. They are given in Table 1. The two parameters being tuned are the number of neighbors $n$ and the distance threshold $t$. The scan fraction is kept low ($f = 0.0005$) because of the size of the training image. Note that if the distance does not reach the threshold during the scan, the best candidate is chosen, and the cell is flagged. At the end of the simulation, all the flagged points are re-simulated once.

SGS and ordinary kriging are applied using the same variogram model presented in Sect. 2.3, and the data follow the same de-trending process. The simulations and the kriging estimate are conditioned using the synthetic GPR lines. Two SGS sets are generated: the set SGS 1 uses 24 neighbors while the set SGS 2 uses 40 neighbors. The kriging estimation is realized using 24 neighbors.

Once the simulations are performed, quality indicators are computed from the predicted and actual topography.

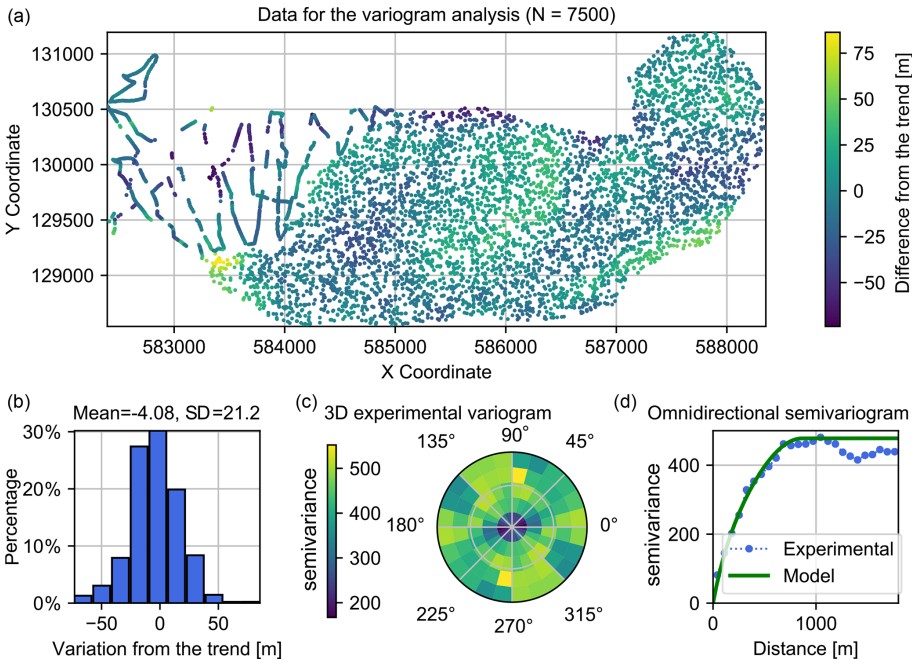

**Figure 3.** Variogram analysis. **(a)** The data set is composed of 7500 points and is assumed representative of the spatial variability of the basal topography. **(b)** The data set has an approximate Gaussian distribution centered around −4.08 m, and no transformation is applied to it. From the 2D experimental variogram map **(c)**, a 2D omnidirectional isotropic model is adjusted against the experimental one **(d)**.

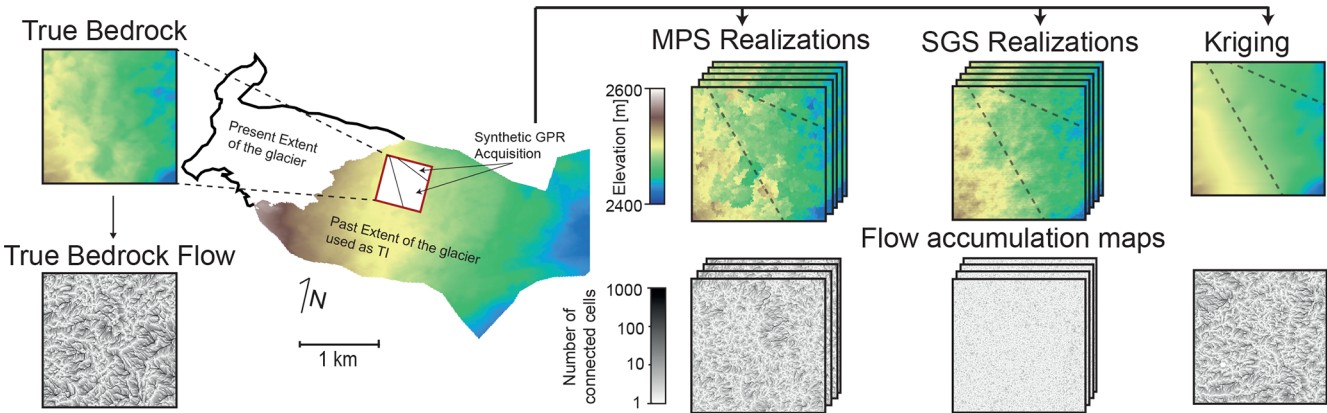

**Figure 4.** The approach used for the systematic tests. Test zones are extracted from the exposed glacier bedrock. GPR lines are also extracted from these zones and used to constrain the geostatistical simulations. It is then possible to compare the different sets with the actual topography. Flow accumulations maps are calculated from the simulated topography and from the reference extracted DEM for comparison.

## 2.5 Quality indicators

We expect that the geostatistical methods that are used to interpolate the basal surface perform differently depending on which derived quantities from the simulated bedrock we compare them to. We compare the fidelity of the different DEMs by evaluating different performance metrics. To illustrate this idea, three quality indicators were designed, based on different uses of the basal interpolated bed. They are defined related to the estimation of the topography itself, the estimation of the overall ice volume between the topography

and the top of an arbitrary ice cover, and finally the estimation of flow accumulation on the basal surface.

A further aspect is that we wish to evaluate both the average match between the forecast and the reference and the predicted uncertainty range. This is why we consider not only the absolute bias but also the continuous ranked probability score (CRPS) (Gneiting et al., 2007) to compare the ensemble of simulated DEMs and ice volumes with the reference.

While the bias quantifies the mismatch between the expected value and the true value, CRPS compares a single true

**Table 1.** Parameter sets used for the synthetic tests.

| Set number | 0 | 1 | 2 | 3 | 4 |
|---|---|---|---|---|---|
| $n$ | 48 | 48 | 48 | 24 | 24 |
| $t$ | 0.01 | 0.1 | 0.05 | 0.01 | 0.1 |

| Set number | 5 | 6 | 7 | 8 |
|---|---|---|---|---|
| $n$ | 24 | 12 | 12 | 12 |
| $t$ | 0.05 | 0.01 | 0.1 | 0.05 |

value $x$ with a cumulative distribution function (CDF) $F$:

$$\text{CRPS}\,(F, x) = \int_{-\infty}^{+\infty} (F(y) - \mathbb{1}(y > x))^2 \mathrm{d}x,$$

where $\mathbb{1}(y > x) = 1$ if $y > x$ and 0 otherwise. CRPS is equal to zero when the prediction is deterministic and equal to the true value. When the prediction displays a sharp distribution centered around the reference, the CRPS is small. If the prediction displays a broader distribution or if the reference is out of the range of predicted values, the CRPS will be larger.

In the following subsections, we define the scores that were applied to the synthetic test cases. Note that to form final scores, each presented score is averaged over all 20 synthetic test cases.

### 2.5.1 Ice volume comparison

For all the test cases, we define the ice volume by fixing (arbitrarily) the altitude $z_c$ of the ice cap to the maximum altitude of the corresponding zone plus 4 m. The reference DEM is referred to as $z_{ij}^{\text{true}}$, with $i = 1, \dots, I$ and $j = 1, \dots, J$ coordinates (spatial indices). The ensemble of the simulated topographies is denoted $z_{ij}^s$, where $s = 1, \dots, S$ is the simulation index, and $S$ is the number of simulations.

The volume of ice $V_s$ for a given simulation is computed as the sum of the differences in altitude between the ice cap altitude and the simulated topography times the area $A$ covered by a grid cell (resolution):

$$V_s = \sum_{i,j} \left( z_{ij}^s - z_c \right) \times A. \qquad (4)$$

The mean value of $V_s$ over the ensemble of $N_s$ simulations is an estimate of the expected value for the volume. We then compute the bias of the estimated value as the absolute difference between the mean and reference volumes. To simplify the computations and facilitate comparisons, the absolute value of the bias for the ice volume is considered per unit area. The elevation of the glacier surface is known; therefore the absolute bias on the volume estimate depends only on the basal elevation simulations:

$$\text{AB}\,(\text{volume}) = \frac{1}{IJ} \left| \sum_{i,j} \left[ \left( \sum_s \frac{z_{ij}^s}{N_s} \right) - z_{ij}^{\text{true}} \right] \right|. \qquad (5)$$

Following the same logic, the CRPS score of the volume prediction is computed by first summing the elevation values in the domain $y_s = \sum_{ij} z_{ij}^s$ for each simulation. The CDF of these $y_s$ values is denoted $Y$, and $y^{\text{true}} = \sum_{ij} z_{ij}^{\text{true}}$. Then the CRPS of the volume estimation is given by

$$\text{CRPS}\,(\text{volume}) = \frac{1}{IJ} \text{CRPS}\,(Y, y^{\text{true}}). \qquad (6)$$

### 2.5.2 DEM comparison

To test if the altitude is properly estimated in all locations of the domain, we compute the absolute value of the bias at each location: it is defined as the absolute difference between the mean DEM (ensemble average over all the simulations) and the reference. To form a single score, this map is averaged over all points in the domain. The mean absolute bias (MAB) for the DEM estimations is therefore defined as follows:

$$\text{MAB}\,(\text{DEM}) = \frac{1}{IJ} \sum_{i,j} \left| \left( \sum_s \frac{z_{ij}^s}{N_s} \right) - z_{ij}^{\text{true}} \right|. \qquad (7)$$

The units of MAB(DEM) and AB(volume) are identical. This allows us to directly compare ice volume errors with DEM errors. Furthermore, MAB(DEM) provides an upper bound for AB(volume).

To compute the CRPS score, let us consider that at each point with indices $(i, j)$ a geostatistical method predicts a distribution of elevation values. The set $\{z_{ij}^1, z_{ij}^2, \dots z_{ij}^S\}$ contains samples drawn from this distribution. Let $Z_{ij}$ be its cumulative distribution function (CDF). It can be approximated by these samples. The mean CRPS of DEM (averaged over all locations) is given by

$$\text{CRPS}\,(\text{DEM}) = \frac{1}{IJ} \sum_{i,j} \text{CRPS}\,(Z_{ij}, z_{ij}^{\text{true}}). \qquad (8)$$

### 2.5.3 Flow accumulation comparison

Flow accumulation is considered here because DEMs are often used to make predictions that are highly affected by geomorphological structures or roughness. Reliably predicting the geomorphological patterns of a DEM is critical, for example, to estimate the velocity at which the glacier may move or to simulate how meltwater can be channelized at the base of the glacier. Flow accumulation maps are thus used in this study because they can be computed rapidly and easily. More importantly, they illustrate the concept of complex geomorphological and roughness description. A flow accumulation map represents the number of cells in each cell that are located upstream and are used to estimate flow direction and catchment delineation (e.g., Tarboton et al., 1991; MacKie et al., 2020b). They are computed by first estimating the flow direction from the local gradients of altitude and integrating the number of cells along the flow path. A very bumpy topography, with many local minima, will lead

to small values of accumulation. A smooth topography will lead to more continuous paths and higher accumulation values. The accumulation is calculated using the Pysheds open-source code (https://github.com/mdbartos/pysheds, last access: 7 August 2021) for watershed delineation. This method is not used to precisely represent hydrological flow at the base of the glacier. Rather, it is used to easily and rapidly compare the results of the application of a highly non-linear process to different bedrock simulations. The ice pressure or the ice coverage is not taken into account, and only the basal topography is used to estimate the cells' connectivity and the possible flow path.

To quantify these differences, the probability density function (PDF) of the flow accumulations for each individual simulation is compared with the PDF of the reference. A standard indicator to compare two PDFs $P$ and $Q$ is the Jensen–Shannon divergence (JSD). It is defined as follows:

$$\text{JSD}(P||Q) = \text{KLD}(P||M)/2 + \text{KLD}(Q||M)/2, \qquad (9)$$

with $M = (P + Q)/2$ and KLD representing the Kullback–Leibler divergence (e.g., MacKay, 2007).

Supposing that the flow accumulation map is given by $f_{ij}^{\text{true}}$ for the reference and by $f_{ij}^{s}$ for the simulation $s$, the probability distributions are created from the flow maps in the following way. First, the set of $K = 14$ bins are defined: $\{B_1, \ldots, B_K\}$: $B_1 = (0, 1]$ and $B_k = (2^{k-2}, 2^{k-1}]$ for $k \in \{2, \ldots, K\}$. Each bin is an interval. We then define $F_k$, the counts of elements $f_{ij}$ which fall into the interval $B_k$, that is

$$F_k = \sum_{ij} \mathbf{1}(F_{ij} \in B_k), \qquad (10)$$

with $\mathbf{1}(F_{ij} \in B_k) = 1$ if $F_{ij} \in B_k$ and 0 otherwise. The counts $F_k$ are normalized so that $\tilde{F}_k$ satisfies $\sum_k \tilde{F}_k |B_k| = 1$, where $|B_k|$ is the length of interval $B_k$. The function $F : \mathbf{R} \to \mathbf{R}$, defined by $F(x) = \tilde{F}_k$ if $x \in B_k$ and 0 otherwise, is a probability distribution function.

Let us call $F^{\text{true}}$ such a PDF constructed from the reference flow accumulation map and $\{F^s, s = 1, \ldots, S\}$ the family of PDFs constructed from the simulated DEMs. Then, the mean Jensen–Shannon divergence (JSD) of flow accumulation is given by

$$\text{MJSD} = \sum_{s} \text{JSD}(F^s, F^{\text{true}}). \qquad (11)$$

## 2.6 Tsanfleuron and Scex Rouge's basal topography estimation

The last step consists of applying the MPS and SGS simulation methods as well as kriging to the actual data set below the Tsanfleuron and Scex Rouge glaciers. The conditioning data are identical for the three methods: the GPR data below the glacier and the DEM around the glacier to ensure the continuity between the border of the simulated area and the exposed altitude of the glacier.

For the MPS simulations, we use the parameters and setup described in detail in Sect. 2.2. We activate the multi-resolution option (Gaussian pyramids) and the relative pattern search, and we use the topography gradient as a secondary variable. This time, only three sets of parameters are used: the ones producing the best scores during the systematic comparison.

The training data set uses the complete exposed part of the glacier's bedrock as a primary variable and its computed gradient as a secondary variable. The DEM is also used directly as hard conditioning data. In total, 120 MPS simulations (40 simulations per parameter set) are generated.

For the sequential Gaussian simulations and the kriging estimate, the variogram that is employed is the one introduced earlier in Sect. 2.3. Both methods use a search ellipsoid of 1500 m and 24 neighbors for the conditioning. In total 40 SGS simulations are generated.

The ice volumes are then computed between all the simulated basal surfaces and the ice topography measured in August (end of summer) 2011 and 2019. For 2011, we use the Alti3D DEM from the Swiss Federal Office of Topography and our DEM for 2019.

For 2019, we use the DEM that was acquired in this work (see Sect. 2.1). The statistics of the volumes are then computed for each method. The standard deviation of the simulated values is used to provide the uncertainty range ($2\sigma$) of the volume. This uncertainty range takes into account only the uncertainty resulting from the spatial variability of the bedrock and its interpolation using an incomplete data set. These estimations do not encompass the other possible sources of error/uncertainty that can arise from the GPR data acquisition (picking, time-to-depth conversion) or for the DEM acquisition. Finally, the flow accumulations are computed for all the simulations.

## 3 Results of the systematic comparison

Figure 5 shows a 3D perspective view of the topographies obtained from one of the 20 test cases. Figure 5a shows the reference DEM for that case, and Fig. 5b, c, and d show one example of MPS simulation, SGS simulation, and the kriging estimation, respectively. As expected, the kriging estimation produces the smoothest and the SGS the roughest topography. The MPS simulation is characterized by an intermediate roughness. The fact that MPS performs better than SGS was expected in this situation because multi-Gaussian random field models are known to maximize the entropy (Journel and Deutsch, 1993), associated in our case with a state of maximum disorder compatible with the data. MPS reproduces the same low-order statistics as SGS, but it also reproduces higher-order statistics, reducing the entropy but improving the simulation compared to the reference. Compared to the reference, and again, as expected, kriging underestimates the small-scale variations. By contrast, the SGS topog-

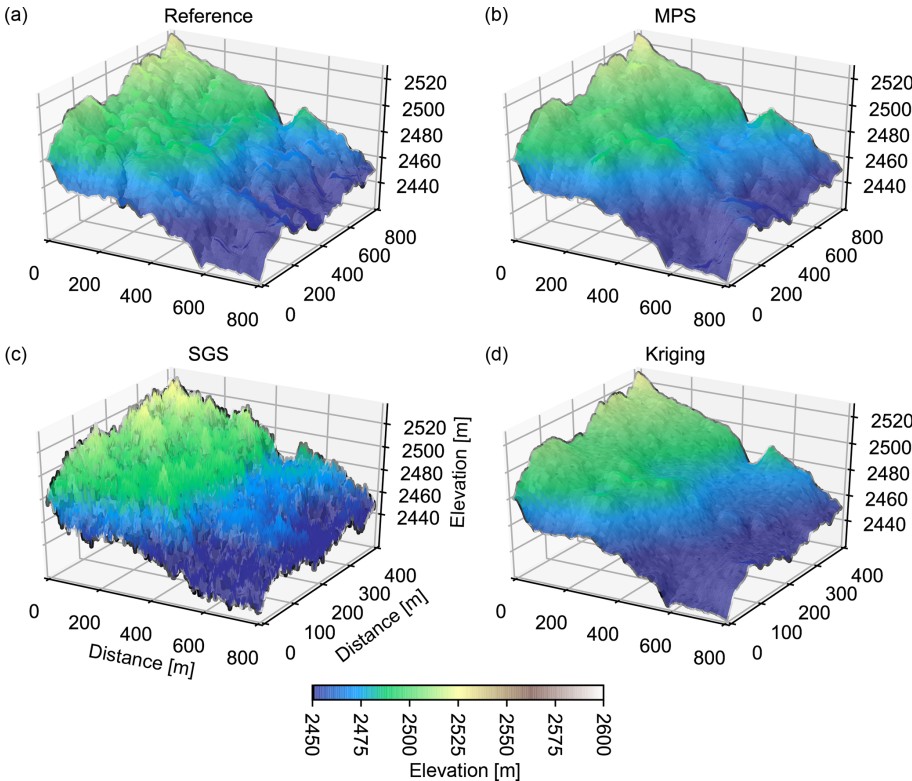

**Figure 5.** A 3D view of topographic interpolation from the synthetic test case. The color corresponds to the simulated elevation. Panel **(a)** presents the reference topography of the test case. **(b)** A MPS simulation, **(c)** a SGS simulation, and **(d)** a corresponding kriging estimate. The roughness is overestimated with the SGS and underestimated with the kriging estimation. MPS simulations provide an acceptable compromise between the method and produce realistic structures.

raphy is rougher than the reference; the complex features of the geomorphology are not properly reproduced even if the large patterns/trends are correctly simulated. The MPS simulation shows more realistic patterns that are continuous and the simulated features appear visually close to the true DEM patterns.

Figure 6 shows the results for four test cases among the 20 that we conducted. The figure is organized as a table. Each row corresponds to one test case. The first column (Fig. 6a) shows the unknown topography for each case, and the black horizontal line indicates the position of the cross section used to display some results in Fig. 6b and c. The last column (Fig. 6d) represents the histograms of volume estimations compared to the reference for each case.

Regarding the cross sections, Fig. 6b shows the results for the MPS technique, and Fig. 6c shows the results for the kriging and SGS methods. The red curve is the reference in all cases. The predictions are represented by the mean and the $2\sigma$ interval estimated from the ensemble of MPS and SGS simulations. The first observation is that the true altitude generally lies very well within the uncertainty range predicted by the different methods. The uncertainty is small on the sides of the domain because the altitude is known at those locations. We observe that the uncertainty is generally larger with

the SGS method than with MPS. This observation can be explained by the fact that even if both methods use the same number of conditioning data, the MPS method is more constrained by the training image and the associated secondary gradient variable. We also observe that the reference topography is sometimes outside of the $2\sigma$ interval, while the SGS distribution always contains the reference. Since the $2\sigma$ interval corresponds roughly to a 95 % confidence interval, it is expected that in 5 % of the simulated cases, the reference altitude should fall outside of the prediction range. Therefore, we argue that the uncertainty estimated with the SGS simulations is too large while the one obtained with MPS is more reasonable. Finally, as expected, the kriging produces a smooth curve that does not represent the small-scale variations in topography but reproduces the mean of the ensemble (the expected value of the altitude in a statistical sense) very well and efficiently and therefore reproduces the general trend of the basal elevation well.

Regarding the volume, Fig. 6d first shows that both the SGS and MPS volume estimations contain the reference within their uncertainty range. The red line always lies inside the histograms predicted by the two methods. The volumes estimated by kriging are, again as expected, close to the mean of the volumes obtained from the SGS simulations. However,

the volumes estimated by kriging can over- or underestimate the reference, and the method does not provide an error estimation. When looking more precisely at Fig. 6d, we observe a higher variability for the SGS method. The MPS volume distributions are better centered around the true volumes. For the SGS volume distributions, the mean is generally further away from the reference than with the MPS method. These observations are confirmed by the CRPS scores of the SGS simulations that are higher than the ones of the MPS simulations (Table 2).

The quality scores presented in Table 2 show that all of the MPS simulations with different parameter sets performed better than the SGS sets for almost all of the indicators calculated. The scores are the closest between the methods regarding the average volume estimated, signifying that on average all the methods perform well to estimate a global volume. However, larger differences between the methods can be observed while looking at the CRPS scores or the pointwise scores. The fact that the MPS performed better in the CRPS score is due to the distributions of volumes that are more precise for MPS than for SGS.

Another important indicator to take into account is the mean Jensen–Shannon divergence of the flow accumulation scores. It reflects how similar distributions of flow accumulation are compared to the reference. The scores of the MPS sets are 10 times better than the score of the SGS and 4 times better than kriging. Figure 7 shows the different flow accumulation probability density distributions for one test case and for the different methods. SGS performs poorly in reproducing the true probability density distribution; it misses the large flow accumulation values. The high entropy of the simulation creates many local minima and cannot reproduce the reference statistical distribution. Kriging surprisingly provides a better distribution in these examples. In general, kriging is known to be smoother and therefore tends to over-represent the large flow accumulation values. For this specific example, the performance of kriging is better than the one of SGS. However, we expect that the kriging performance will depend very much on the spatial density of the data, when SGS will not. The MPS probability density curve is the closest to the true DEM one and implies a good performance in pattern reproduction.

Finally, the systematic tests showed that the best parameter sets were numbers 6, 3, and 8 for MPS and the 24-neighbor parameter set for SGS. These parameters were then used for the practical application of Tsanfleuron Glacier.

## 4   Tsanfleuron Glacier's results

Figure 8 shows the comparison between the basal topography interpolated using SGS, kriging and MPS on a map view and along two cross-sections through the glacier. The mean global shapes produced by the three methods are similar. The SGS simulations tend to be a little noisier, but the mean value shows no significant difference between the methods.

Table 3 provides the estimated ice volume and uncertainties. The results from the three methods are very close: between 113 and 114 million cubic meters for 2019. With kriging, it is not possible to estimate the uncertainty range on the volume, but with the SGS and MPS simulations, we obtain similar uncertainty ranges on the order of 1 % for the volume. For comparison, and to evaluate the ice loss between 2011 and 2019, we also compute the volumes in 2011. The 2011 and 2019 surface DEMs were both acquired in August. According to the simulations, the Tsanfleuron and Scex Rouge glaciers lost about 25 % of their total ice volume in 9 years.

So far, the results obtained by the three methods are very close, because the differences in the spatial distribution of the basal elevation values are compensated for when we integrate them over the whole glacier area to compute the volume or mean altitude.

However, the flow accumulation results (Fig. 9) are very different and highly affected by the detailed geomorphological structures in the interpolated surfaces. The SGS simulations lack realism in this case and cannot predict the high-flow accumulations. They are limited to short paths, with many local minima due to an overestimation of the roughness of the simulated topographies. On the other hand, kriging provides an interpolated topography that is smoother than the reality and overestimates the flow accumulations. Finally, MPS is able to generate more realistic basal topographies. The predicted topographies display geomorphological features and roughness that are similar to the structures visible in the uncovered bedrock area.

## 5   Discussion

### 5.1   MPS parameterization

When comparing the interpolated basal topographies with the three methods, our results show that the MPS approach provides the simulations that display geomorphological features that are the closest to the data set. But to obtain those results, the DeeSse algorithm needs to be properly parameterized, and adequate secondary variables have to be used. During this project, we have tested various options. Using the topography gradient as a secondary variable proved to be a simple and efficient solution, but further improvements could certainly be made. One possibility that we considered but did not implement would be to use as secondary variables two hillshade projections, 45° apart from each other. Adding these two variables would allow us to account both for the absolute steepness of the topography but also the orientation in space of the geomorphological patterns.

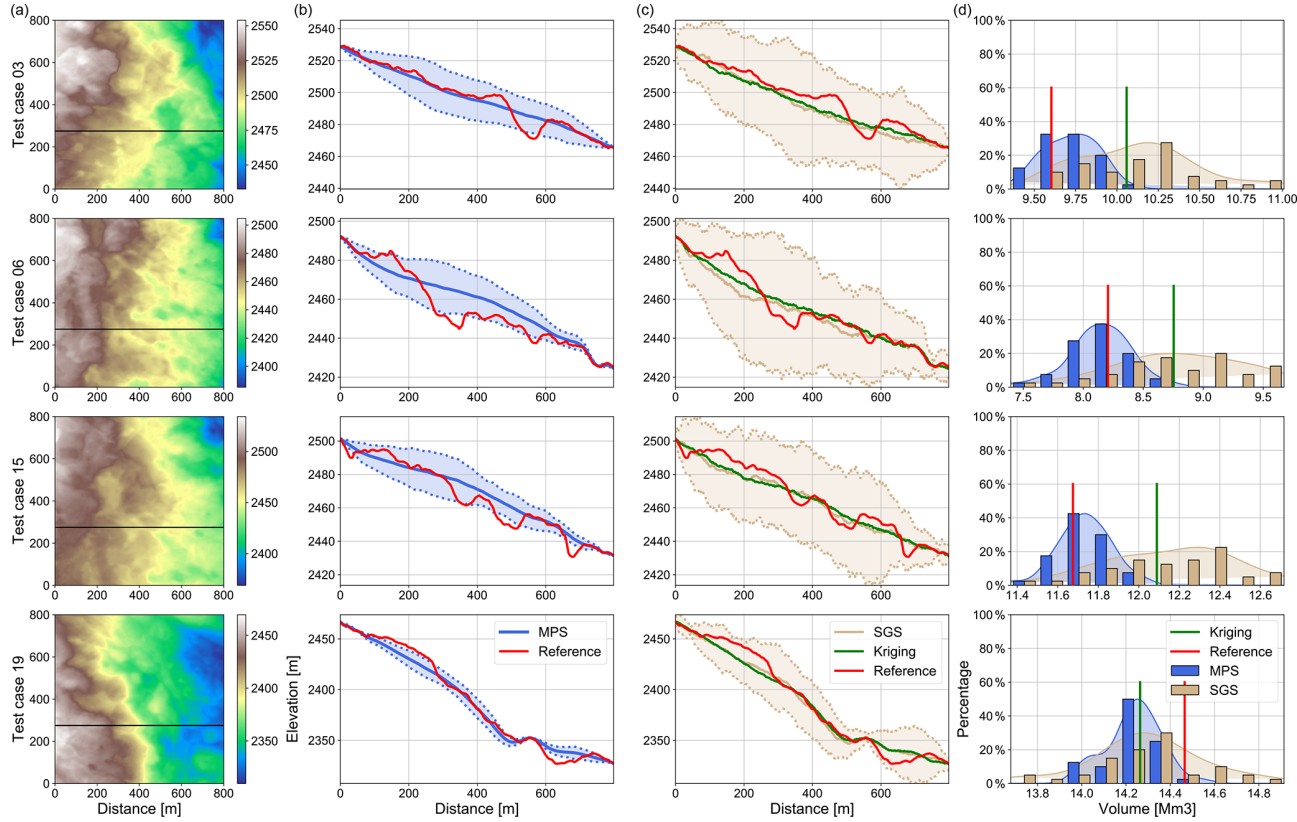

**Figure 6.** Synthetic test results. Panel **(a)** displays the reference DEM in a plan view, with the black line showing the location of the cross sections. Panel **(b)** presents the distribution of the cross section simulated with the MPS method (parameter set 3). Panel **(c)** presents the distribution of the cross section simulated with the SGS methods (parameter set 1). Panel **(d)** presents the different volume distribution of the methods against the reference one in red.

**Table 2.** Quality indicators averaged over all realizations for different simulation methods (MPS, SGS, kriging) and for different parameter sets. The rows are sorted by best (lowest) CRPS score of ice volume estimation. CRPS score of ice volume estimation, mean absolute error of the ice volume, CRPS score of DEM averaged over all points, mean absolute error of DEM averaged over all points, and Jensen–Shannon divergence of flow accumulation distributions averaged over all simulations.

| Method | Set | CRPS (volume) | AB (volume) | CRPS (DEM) | MAB (DEM) | MJSD flow |
|--------|-----|---------------|-------------|------------|-----------|-----------|
| MPS | 6 | 1.41 | 2.01 | 3.59 | 4.96 | 0.014 |
|     | 3 | 1.42 | 1.91 | 3.66 | 4.92 | 0.013 |
|     | 8 | 1.47 | 2.04 | 3.64 | 5.01 | 0.015 |
|     | 7 | 1.53 | 2.11 | 3.68 | 5.07 | 0.015 |
|     | 5 | 1.60 | 2.14 | 3.79 | 5.06 | 0.014 |
|     | 4 | 1.61 | 2.11 | 3.79 | 5.06 | 0.014 |
|     | 0 | 1.62 | 2.06 | 3.88 | 4.98 | 0.018 |
|     | 2 | 1.70 | 2.11 | 3.95 | 5.05 | 0.019 |
|     | 1 | 1.70 | 2.11 | 3.95 | 5.05 | 0.019 |
| SGS | 0 | 1.97 | 2.86 | 4.17 | 5.54 | 0.109 |
|     | 1 | 1.99 | 2.96 | 4.18 | 5.61 | 0.111 |
| Kriging | 0 | 2.91 | 2.91 | 5.45 | 5.45 | 0.047 |

**Table 3.** Volumes of ice computed with the 2019 and 2011 DEMs for the Tsanfleuron and Scex Rouge glaciers.

| Method | Tsanfleuron 2019 [million cubic meters] | Tsanfleuron 2011 [million cubic meters] | Scex-Rouge 2019 [million cubic meters] | Scex-Rouge 2011 [million cubic meters] | Total 2019 [million cubic meters] | Total 2011 [million cubic meters] |
|---|---|---|---|---|---|---|
| MPS | $109.8 \pm 1.5$ | $143.2 \pm 1.5$ | $4.1 \pm 0.3$ | $6.6 \pm 0.4$ | $113.9 \pm 1.6$ | $149.8 \pm 1.6$ |
| SGS | $109.7 \pm 1.7$ | $143.3 \pm 1.8$ | $4.2 \pm 0.2$ | $7.05 \pm 0.2$ | $113.9 \pm 1.8$ | $150.4 \pm 1.8$ |
| Kriging | 109.0 | 143.0 | 4.1 | 6.8 | 113.1 | 149.9 |

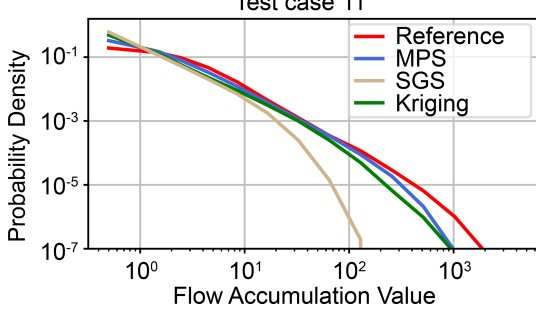

**Figure 7.** Probability distributions of flow accumulation values for example realization 11. The probability distribution of flow accumulation for the true DEM is compared to an example MPS simulation, kriging map, and SGS simulation. The corresponding Jensen–Shannon divergences with respect to the true distribution are 0.015 (MPS), 0.228 (kriging), and 0.107 (SGS).

## 5.2 Ice volume of the Scex Rouge and Tsanfleuron glaciers

The results from the previous section show that the three geostatistical methods are able to provide consistent and comparable estimates of the mean basal topography and the overall ice volume. This similarity is expected for the SGS and kriging because the volume calculation is a linear function of the basal topography, and in this situation kriging and SGS simulations will provide the same mean value (see, e.g., Chiles and Delfiner, 2012). It was not obvious that MPS would also provide the same volume because the underlying spatial variability model is different. Kriging and SGS are based on the same variogram model, but MPS is based on the complete topographic data set and accounts for higher-order statistics. The fact that the three techniques provide a consistent result is therefore an interesting finding. At the glacier scale, the spatial distributions of the thicknesses (Fig. 10) according to the three methods are consistent. The thickest area is in the northern part of the Tsanfleuron area and presents a roughly E–W orientation.

Only the SGS and MPS methods are able to estimate the uncertainties on the total volume. As explained before, even if kriging can, at any point, provide the variance of the altitude, it cannot be used directly to infer uncertainties on the volume. Covariances between any pair of points would need to be considered and integrated over the whole domain. The volume uncertainties estimated with the SGS and multi-Gaussian model are in general reliable (see, e.g., Chiles and Delfiner, 2012). Here, we show that the uncertainties obtained with the MPS are consistent with those obtained with SGS.

For the Scex Rouge and Tsanfleuron glaciers, these three methods allow us to obtain an estimation of $149.8 \pm 1.6$ million cubic meters of ice for 2011. This result is larger than the 100 million cubic meters estimated by Gremaud and Goldscheider (2010) for 2008. We explain this difference by considering that Gremaud and Goldscheider (2010) employed a different geophysical technique and had fewer data points over the glacier to interpolate the ice thickness (using kriging). More precisely, they used a radio magneto-telluric geophysical method; they also inverted the geophysical signal using two layer assumptions, one representing the glacier and the other the underlying limestone. The authors indicate that they may have underestimated the actual thickness of the glacier. It could be due to an erroneous estimation of the electrical resistivity of the glacier possibly being affected by the presence of meltwater. Another geophysical campaign was conducted by Nath Sovik and Huss (2010). They report much deeper ice thickness reaching up to 180 m in the central part of the glacier. Personal communication with these authors indicates that they are uncertain about the very deep data; they obtain a volume on the order of 200 million cubic meters for the Tsanfleuron Glacier only in 2009.

Another indirect comparison with existing data was made using year-to-year mass balance (GLAMOS-Glacier Monitoring Switzerland, 2019). An ice density value of $850 \times 60 \, \text{kg m}^{-3}$ was used as recommended in Huss (2013) to perform the conversion between water equivalent (w.e.) and ice volume. With this method, the loss of volume is estimated to be around $34 \pm 2.5$ million cubic meters of ice between September 2011 and 2019. This value is similar to the one we derived with our approach ($35.9 \pm 3.2$ million cubic meters for the MPS).

To finalize this discussion, we would like to recall that several operators did the picking of the depth of the reflectors independently on our GPR data and that the order in which

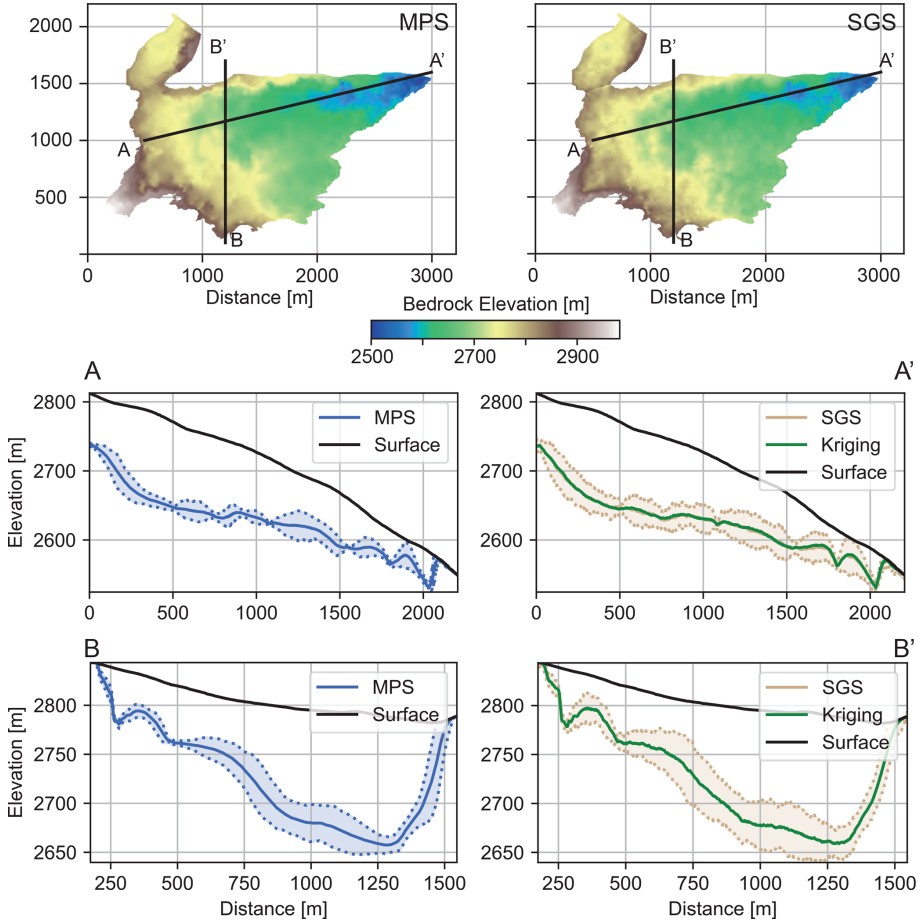

**Figure 8.** Simulated basal topography for the MPS method (in blue) and for the SGS method (in brown). The kriging-estimated topography is displayed alongside the SGS simulation in green.

the data were presented to the operator was randomized. We also compared the result of the picking from the different operators and removed the parts of the data that were not consistent. Even if the surface and volume estimations may still suffer from some remaining errors, we tried to apply these strict procedures to avoid bias, and therefore, we expect that the data set and the estimated volumes are as reliable as possible.

As expressed earlier, the present analysis did not consider the uncertainty regarding the velocity used to convert the two-way travel time data from the GPR to depth. We used a uniform value that corresponds to the wave velocity in cold and non-wet ice generally used. However, Tsanfleuron Glacier is described as a polythermal glacier (Hubbard et al., 2003), and consequently it potentially has a layer of temperate ice with a non-negligible content of liquid water. This is also indicated by the presence of typical scattering in the processed GPR data (see Fig. 1) (Schannwell et al., 2014). Inside this layer, the wave propagation speed should be lower than in the rest of the glacier. Endres et al. (2009) show that an increase of a few per-

cent in water content between the two layers can change the propagation speed from $0.1703 \pm 0.0003\,\mathrm{m\,ns^{-1}}$ down to $0.1629 \pm 0.0014\,\mathrm{m\,ns^{-1}}$, which corresponds to a 5 % variation. The velocity used here lies in between these two values, and while it is certainly not exact and identical for the whole glacier/layers, it can be assumed realistic. Finally, the real time-to-depth conversion factor is non-uniform in space: it depends on the thickness of each layer and is a weighted average of the cold and temperate layer velocities. The spatial distribution of the transition depth of the layers is not known, and therefore an accurate quantification of this uncertainty has been considered as out of the scope of this paper.

## 5.3 Which geostatistical method for which purposes?

To rapidly estimate the volume of ice, our results show that the kriging method provides a value that is reasonable. However, kriging cannot be used to directly obtain the uncertainty on the volume.

We argue that any scientific estimation should always be accompanied by an uncertainty estimation when possible. Hence, it is therefore preferable to directly use the SGS or

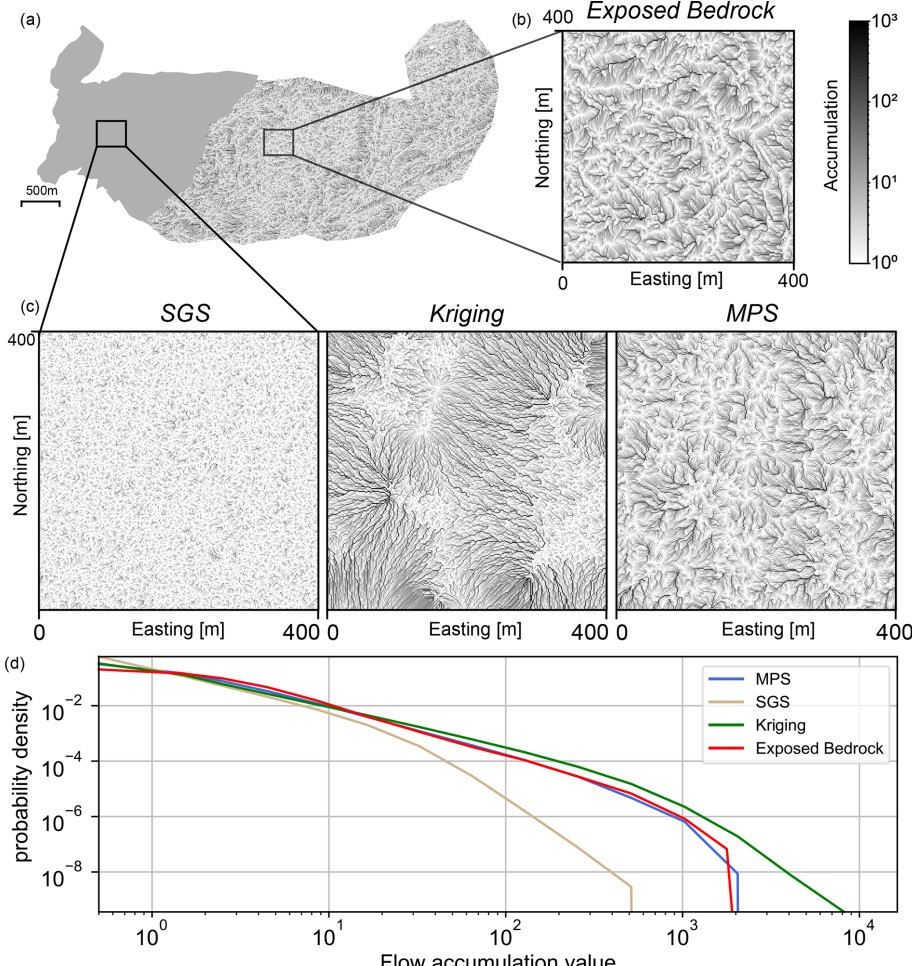

**Figure 9.** Flow accumulations calculated from the three methods and compared against the one computed in the exposed part of the bedrock used as reference and the probability density of accumulation values. The SGS simulation underestimates the length of the connected path, while kriging overestimates it. The visual patterns and the flow accumulation distribution obtained with MPS are the closest to the exposed bedrock reference.

MPS approach to get not only the volume but also the corresponding uncertainty. The two methods provide comparable results. The SGS method requires a variogram model from the experimental variogram of the data. By contrast, the MPS method simply requires providing an exhaustive data set that represents the type of spatial variability that is analog to the patterns that are expected below the glacier. In a previous study (Dagasan et al., 2019), we have shown for a different application that even if the training data are slightly different from the spatial patterns that are actually occurring at the site of interest, it is possible by cross-validation to adjust the parameters of the MPS simulation to compensate for this mismatch and obtain satisfying results.

Finally, if the estimated topographies of the bedrock below the glacier have to be used to estimate a quantity that derives non-linearly from the topography, the MPS method should be used. Indeed, we have shown that MPS provides a much better reproduction of the geomorphology of the simu-

lated basal surfaces: the results are much closer to the reference than the other two techniques. This result confirms the observations made by MacKie et al. (2020a) concerning the importance of using MPS for the estimation of the presence of underglacial lakes. The importance of accurately simulating the roughness of subglacial topography was also already highlighted by Goff et al. (2014), who used a combination of multi-Gaussian simulations with deterministic trends, but the procedure that we propose with MPS is simpler to implement.

## 6   Conclusions

This study presents an example of the benefits of using advanced geostatistical methods for basal topography interpolation and compares three methods: kriging, sequential Gaussian simulations, and multiple-point statistics.

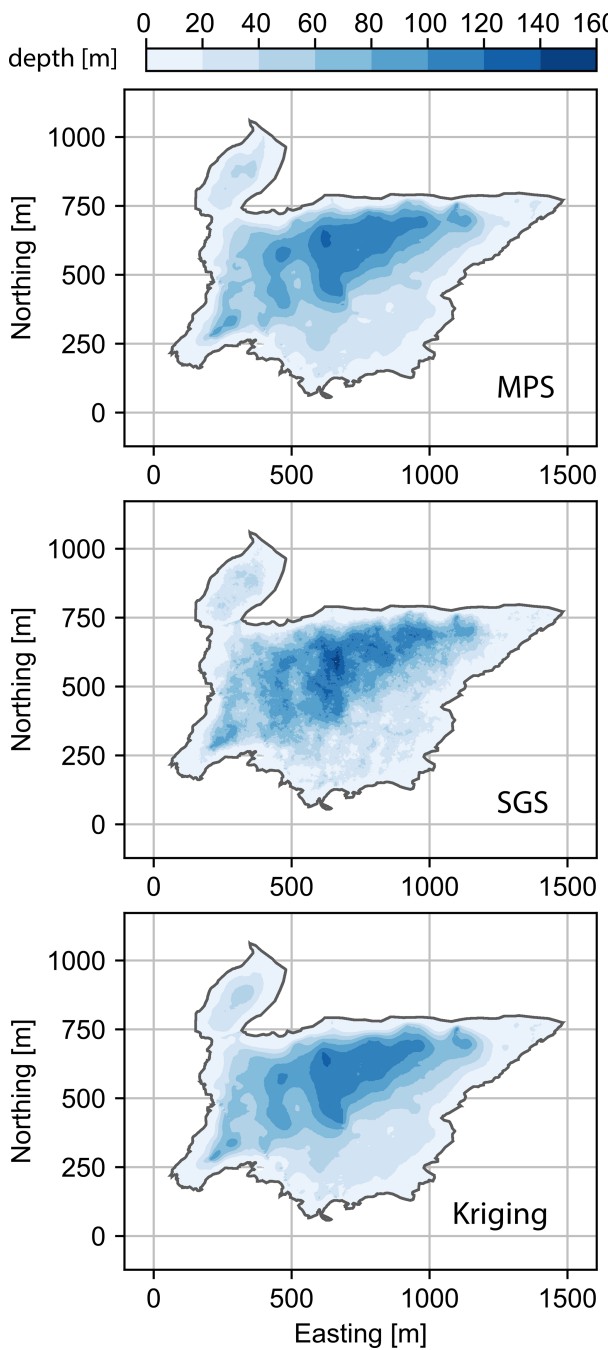

**Figure 10.** Ice thickness calculated from the three methods using the 2019 surface DEM. The SGS and the MPS basal models used are the averaged model over all the realizations.

The three methods are able to provide consistent and similar volume estimations. Kriging and SGS require the analysis of the statistics of the data, adjust a spatial trend, and identify a variogram model. Once this is done, these two methods provide local estimation of uncertainty. But to get the uncertainty on the volume, one needs to use the SGS method.

The MPS technique is the most versatile: it provides a comprehensive estimation of the volume, as well as local uncertainties comparable with the other methods. But in addition, it is able to produce realistic basal structures, even in areas where the data are scarce or the structure complex. Compared to MPS, SGS and kriging tend to produce interpolated surfaces that are respectively too smooth or too noisy. Therefore, they can lead to biased predictions when they are used to derive quantities that depend strongly on the detailed geomorphological structures of the basal topography as illustrated with the flow accumulation calculations done in this paper. The same types of errors are expected if these topographies are used to simulate the glacier movement or the basal flow of melted water and the channeling of this flux. In these situations, the detailed structures may be even more important than the global trend, and the MPS approach is recommended. The main limitation of the MPS approach is that it requires a TI that is representative of the glacier basal structures. Finding the proper analog data is therefore an important part of the approach and may be difficult for large glaciers with little information about the underlying geology. In these situations, a possibility could be to test various training images using $k$-fold cross-validation techniques as shown in Juda et al. (2020). The case of steep valleys where sediments cover the bedrock can also make the TI selection complex.

Based on the results obtained in this paper and those published by MacKie et al. (2020b), we suggest that the MPS approach could help enhance the accuracy of glacier evolution models. The MPS simulations can be used to better define the initial conditions as well as to provide a more realistic basal geometry (boundary condition). In addition, because the method is able to generate a set of simulated topographies, it could be used as the base for a Monte Carlo analysis. Each simulation can be used as input for one run of a glacier evolution model. By analyzing the ensemble of results, one can quantify the uncertainty on glacier evolution related to an imperfect identification of the basal geometry of the glacier. Looking even further, and making an analogy with how the MPS approach is used for groundwater applications (e.g., Jäggli et al., 2018), we can imagine coupling the MPS approach and glacier evolution model in an inverse Bayesian modeling framework to infer a plausible basal geometry when little glacier depth data are available and one has to rely on indirect surface observations.

Finally, the ice volumes calculated for the Scex Rouge and Tsanfleuron glaciers with the three methods are in accordance with the mass balance calculation and are linked with robust error estimation. Our results indicate that there

has been a significant mass loss at this glacier and that these methods enable higher-accuracy ice loss estimates and could enable improvements in glacier retreat projections. Such improvements could be important for global awareness, political decisions, and preparing the mountains' infrastructure for its future evolution.

*Data availability.* A simulation obtained with each method, the mean simulations of SGS and MPS, and the DEM and the conditioning point sets are available at https://github.com/randlab/tsanfleuron_glacier_data (last access: 24 October 2021, https://doi.org/10.5281/zenodo.5675737, Neven, 2021).

*Author contributions.* AN coordinated and conducted a part of the fieldwork. AN processed the GPR and drone data. AN, VD, JS, and PR designed and tested the different geostatistical procedures. AN and VD prepared the data. PJ participated in the data acquisition and designed and implemented the quality indicators. AN, VD, and PJ wrote the paper. PR initiated and supervised the work, conducted the field acquisition, and was involved in the writing and editing of the paper.

*Competing interests.* The contact author has declared that neither they nor their co-authors have any competing interests.

*Acknowledgements.* The authors would like to thank Marie Vallat and Cyprien Louis, two students who participated in the GPR data collection and initial geostatistical analysis. The authors are also thankful to James Irving at the University of Lausanne, who provided the GPR equipment and support. They would like to thanks the Prarochet Hut and the "Glacier 3000" company for the logistic support. Finally, they want to thank the editor Adam Booth, the anonymous reviewer, Clemens Schannwell, and Emma MacKie for their numerous comments that helped improve the quality of the paper.

*Review statement.* This paper was edited by Adam Booth and reviewed by Clemens Schannwell, Emma MacKie, and one anonymous referee.

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
