# Peer review of "Ice volume and basal topography estimation using geostatistical methods and GPR measurements: Application to the Tsanfleuron and Scex Rouge Glacier, Swiss Alps"

_The Cryosphere, 2021_

## Referee Comment (RC2)

**Review of Neven et al. "Ice volume and basal topography estimation using geostatistical methods and GPR measurements: Application on the Tsanfleuron and Scex Rouge glacier, Swiss Alps""**

August 6, 2021

**General comments:**

In the manuscript presented by Neven et al. three different geostatistical methods are compared for a small mountain glacier in the Bernese Alps, Switzerland. The authors use first a synthetic test case to calibrate the free parameters of the different models. The three different methods are then applied to the Tsanfleuron and Scex glacier to compare the resulting ice volume and basal topography between the three methods. They conclude that while for integrated quantitites such as ice volume all methods give very similar results, the presented multiple points statistics method should be the method of choice for interpolation of basal topography.

As a non-expert of geostatistical interpolation methods, I found the comparison between the three methods quite interesting and think the presented material is appropriate for a journal like TC. The Figures are in most cases quite nice, but the text certainly needs polishing. I found quite a number of typos and the text sounds a little awkward in places (see comments below).

I think the focus of the paper should almost entirely be on the methods comparison. I am much less convinced of what the authors call the applied section of the paper (e.g. the magnitude of the ice volume and the future "projection" based on the past ice volumes). This is mainly because of some assumptions that were made in the GPR data processing that are only very briefly mentioned (see main concerns below). I therefore recommend to be a bit more cautious in the interpretation of the ice volume numbers. As mentioned above,

I am certainly not an expert for geostatistical methods, which means I cannot comment too much on the MPS or SGS algorithms. But I think there a many instances in the manuscript where a little less technical language could make the paper more accessible to a broader audience.

I hope the authors find my comments below helpful for the revision of their manuscript.

**Specific comments:**

**Main concerns:**

1. In my view, the authors should be very careful in interpreting their absolute values of ice volume that they get from all their methods. The primary reason why I am so sceptical is that in L119 the authors say that they used a uniform wave propagation speed (0.168 m/ns) for the time-depth conversion. This is a value that is commonly used for cold ice. However, Tsanfleuron glacier is a polythermal glacier (Hubbard et al. 2003) which means there is a temperate ice layer of significant depth (Schannwell et al. 2014), where this wave propagation speed is certainly lower than what is assumed. While this assumption is absolutely fine for the comparison of the three methods, because the same assumption is used for all of them, it becomes problematic when the absolute ice volume is interpreted. I therefore would like to see this discussed in more depth and add a few sentences why the presented numbers might be off.

2. I have a few general comments/questions about some aspects of the geostatistical methods that I feel would help to gauge how applicable (or not) this method is for other ice masses.

   - Can you comment on how sensitive the MPS is to the selection of your TI?

   - Given the fact that the TI needs to have similar structures to what is expected under the glacier. How restrictive is this assumption? I guess this is a valid assumption for a small mountain glacier, if the lithology does not change? Could this be used for ice sheets?

   - Just to clarify, where ever you have GPR measurements, the interpolated value corresponds to the measured value exactly?

   - In Figures 6 and 8, the mean of basal topography for MPS and Kriging look pretty much identical to me. Is this just because for these lines they are not different or do I get almost the Kriging topography if I average over all MPS simulations? If I do get the same topography, does that mean that the only difference between the two is MPS comes with uncertainty bounds and Kriging doesn't?

3. I think the "Conclusions" section needs rewriting. Certainly scratch the first paragraph. In its present form, there seems to be a bunch of different ideas just listed one after the other. My suggestion would be to really highlight the important points:

- You compare different geostatistical methods

- Why (and in what situations) is MPS best? What is the drawback of the method?

- Then highlight where this method could be applied (e.g. boundary condition for glacier models, glacial geomorphology etc.)

**Technical corrections:**

Title:
Would change "Application on" to "Application to"

Abstract:
L1 delete nowadays
L1 maybe change to "mountain glacier thickness" as airborne methods are predominately used for ice sheets?
L5 I did not know what conditioning data was at first reading. Is there a more accessible term for this?
L11 check formatting throughout. Should be Mio m$^3$
L13 here and throughout change under-glacial to subglacial

L17 This statement could use a citation

L24 I would delete "nowadays"

L25 Would change to "thickness of smaller ice masses"

L28-29 This sentence could use a citation

L29 "generally used are unable to provide"

L40 A bit often "Furthermore". Would recommend to delete this one or the previous one.

L42 "cannot succeed"

L49 "subglacial/or basal topography"

L55 "infers it implicitly from"

L69 "where the target topography is known. This highlights the advantages"

L74 "but are located"

L77 "According to the"

L78 "applied to all Swiss glaciers"

L77-80 This relates to my main concern from above. So based on the presented numbers from 2009 and 2016, the volume has doubled in 7 years. Which of the numbers is more trustworthy? Especially given the fact that your numbers lie pretty much in between these estimates. Why did they have problems with the picking and why did you not have these problems?

L87 "original algorithm. In parcticular DeeSse"

L97 "numerical experiment was designed"

L101 I do not fully follow how you compare absolute volume for a non-glaciated area? I think you say later that you set the surface elevation to 4 m, regardless of the basal topography. Is this correct? Could you make this more clear?

L124 "with a smaller signal-to-noise ratio. An example"

L131 Is there a reference for the Pix4D software? Is is open source?

L135 Does the choice of a random path affect the final result? Would the results be very different if I chose a more regular pattern?

L157 "along altitude"

L184 Is there a reference for the Arc2gems software? Is is open source?

L189-190 This sounds awkward. Does this mean that you add some white noise on top of the bed elevation data? If so, please reformulate.

L203 Can you give some examples of quantities that can be predicted and which cannot?

L205 "on average"

L250 "Reliably predicting the geomorphological"

L276 "consists of applying"

L292 "each method"

L294-296 As pointed out above, it is good that you mention it here that other sources of uncertainty are not included. However, you should make sure that you also keep that in mind when you interpret your ice volume estimates. In addition to your assumption that there is no water in the ice. You also implicitly assume that the thickness of the temperate ice layer did not change between 2011 and 2019, which is rather unlikely (see Gusmeroli et al. 2012). I do not see a straightforward way to incorporate this, but the reader should be made aware of this.

L298-308 Here and throughout, please make sure that you use the same formatting and spelling when you refer to Figures in the text.

L328 "is generally further away from the reference"

L329 I think it should be "higher than the ones of the MPS"

L365 I think this could be mentioned as a limitation of the MPS method more clearly. It can only work if structures between the TI and what is interpolated are similar. That is not surprising as the algorithm tries to match patterns from the TI.

L367 "MPS parameterization"

L376 "section show that the three"

L378 "kriging and SGS simulations will provide"

L386 "multi-Gaussian"

L391 "had fewer data points"

L411 "sources is surface topography of the glacier"

L411-413 Again, I think this is absolutely fine, but please interpret the absolute numbers with this in mind as well.

L417 More of a comment than anything really. I agree in a perfect world we would always like to have an uncertainty estimation, but I think it is also worth to keep in mind that some scientific estimates/simulations are way too computationally expensive to reliably quantify uncertainties.

L426 "Indeed"

L426 "importance of accurately simulating the roughness"

L426 delete "somehow"

**Figures:**

Fig. 1: Why are there white stripes in the aerial image?

Fig. 2: In Figure 2 lower panel, you can see quite well the temperate ice (scattered) layer

Fig. 3: Hard to tell whether "1/2 (Z(x) ..." is the label of the colourbar in (c) or the label of the y-axis in (d).

Fig. 6: To me it looks like kriging has more short-wave variability than the MPS mean. Is there an explanation why MPS is struggling with Test case 19?

Fig. 7: I'm not sure but shouldn't the SGS value be higher here?

Fig. 8: Differences in the upper panel are really difficult to see. Maybe you could instead show a difference plot? And why is the kriging topography not shown?
In the caption: "is displayed alongside the SGS"

Sincerely,
Clemens Schannwell

**References**

Gusmeroli, A, Jansson, P, Pettersson, R and Pettersson, R (2012) Twenty years of cold surface layer thinning at Storglaciären, sub-Arctic Sweden, 1989–2009. J. Glaciol., 58(207), 3–10

Hubbard, BP Hubbard, A, Mader, HM Tison, JL Grust, K and Nienow, PW (2003) Spatial variability in the water content and rheology of temperate glaciers: Glacier de Tsanfleuron, Switzerland. Ann. Glaciol., 37, 1–6

Schannwell, C., Murray, T., Kulessa, B., Gusmeroli, A., Saintenoy, A., Jansson, P. (2014). An automatic approach to delineate the cold–temperate transition surface with ground-penetrating radar on polythermal glaciers. Annals of Glaciology, 55(67), 89-96.

---

## Referee Comment (RC3)

**Review of "Ice volume and basal topography estimation using geostatistical methods and GPR measurements: Application on the Tsanfleuron and Scex Rouge glacier, Swiss Alps" by Neven et al.**

**General comments:**

This study demonstrates different topographic interpolation techniques for two glaciers in the Swiss Alps and investigates their effect on a subglacial water routing model. The authors interpolate GPR measurements of bed topography using kriging, sequential Gaussian simulation (SGS), and multiple-point simulation (MPS) for the Tsanfleuron and Scex Rouge glaciers, as well as synthetic examples. MPS is implemented with a secondary variable, the bed gradient. The authors test different simulation parameters to determine the optimal model setup. The ice volume and hydrological flow paths are computed for each bed elevation model. The authors conclude that MPS is the most robust method for interpolating subglacial topography.

Overall, this study is clear and rigorous and addresses an important problem in glaciology. This study provides a thoughtful and thorough comparison of different interpolation methods that TC readers will find interesting. The authors demonstrate novel interpolation methods and performance metrics that are relevant to topographic interpolation, ice volume estimation, and subglacial hydrology.

Some figures require minor changes, and there are some typos and grammatical issues (see line comments below). There are some paragraphs that are only one sentence long. I recommend appending these sentences to other paragraphs. It would be helpful for general glaciology audiences to define terms such as conditional simulation, variogram, hard data, and non-stationarity. I also recommend providing a brief overview of variograms and the SGS methodology in Section 2.3. While SGS is a well-established method, TC audiences will benefit from an explanation.

**Main concerns:**

The authors state that kriging cannot be used to compute ice volume uncertainty, but that is not the case. While it is true that kriging does not sample the uncertainty space in the same way that simulation does, kriging can be used to provide the variance or standard deviation of an estimate at any given location. In theory, multiple realizations produced by sequential Gaussian simulation should converge to a distribution that is represented by the kriging solution. For completeness, I recommend that the authors compute the uncertainty in ice volume for the kriging interpolation.

The authors fit a polynomial trend to the data in order to perform the kriging interpolation and sequential Gaussian simulation. This is a somewhat arbitrary, but often necessary, step for variogram-based methods. How was the degree of the polynomial chosen? Does it matter for your results that the synthetic examples are not detrended? I would like to see more justification for the polynomial selection and discussion on the implications of trend estimation. I also recommend mentioning in the discussion that MPS does not require trend estimation, which is another advantage of the MPS method.

More information is needed on the hydrological modeling method. The Pysheds package that the authors use only has examples for topography without ice. The authors do not state whether or not they account for ice overburden pressure in their hydrological modeling. If that is difficult to do with Pysheds, Chad Greene has a nice tutorial (https://www.mathworks.com/matlabcentral/fileexchange/55352-how-to-estimate-subglacial-water-routes). The synthetic examples have a flat ice surface, which could bias the hydrological models. More justification is needed for using the flow accumulation values for the synthetic examples.

I would like to see more discussion on the implications of the hydrological findings. The authors compare the distributions of flow accumulation values for different interpolation methods, but it is unclear why this matters. Perhaps flow accumulation is important for discriminating between a channelized or distributed drainage system? It may also be helpful to refer to studies by Zuo et al., (2020) and MacKie et al., (2021) which previously investigated the impact of MPS and SGS on hydrological flow.

Zuo, C., Yin, Z., Pan, Z., MacKie, E. J., & Caers, J. (2020). A Tree‑Based Direct Sampling Method for Stochastic Surface and Subsurface Hydrological Modeling. *Water Resources Research*, *56*(2), e2019WR026130.

MacKie, E. J., Schroeder, D. M., Zuo, C., Yin, Z., & Caers, J. (2021). Stochastic modeling of subglacial topography exposes uncertainty in water routing at Jakobshavn Glacier. *Journal of Glaciology*, *67*(261), 75-83.

**Line comments:**

Lines 13-14: "significantly improve for example the precision of under-glacial flow estimation" For clarity, specify that you are referring to hydrological flow. Add commas before and after "for example"

Line 19: no comma needed after "crucial"

Line 28: "Depending of" should be "Depending on"

Lines 30-32: "the choice of the method becomes critical since the flow process is highly non-linear and is strongly linked to the morphology of the subglacial topography"

I recommend clarifying that you are referring to hydrological flow, not ice flow (which is also non-linear and dependent on morphology). What is meant by a non-linear flow process? I think the authors mean that flow accumulation is not a linear function of bed elevation.

Line 35: "produces by construction"

Awkward wording. It would be sufficient to just say "produces"

Lines 36-38: "Furthermore, even if kriging allows estimation of the local uncertainty on the elevation of the bedrock, it cannot be used to estimate the uncertainty of the global volume of ice (see e.g. Chiles and Delfiner, 2012, p. 478)."

Why can't the uncertainties from kriging be used to estimate ice volume uncertainty? What happens when you use the kriging bed uncertainties to estimate ice volume uncertainty?

The reference to Chiles and Delfiner (2012, p. 478) does not support this statement. Chiles and Delfiner (2012, p. 478) describe a scenario where surface area increases with roughness. It is true that kriging underestimates the surface area of topography (if each grid cell is represented by a tilted plane), but this shouldn't affect the volume calculation.

Line 41: "two points spatial statistics" should be "two point spatial statistics"

I recommend elaborating on this sentence so that this concept is more understandable to non-geostatisticians. It might be more understandable to say that these methods are based on the variance between pairs of points, and briefly state what a variogram is.

Line 51: "that the one" should be "than the one"

Line 55: "require to define" → "require the definition of"

Lines 54-55: "MPS does not require to define an analytical two-point statistics model to represent the spatial variability but instead infers it in an implicit way"

MPS does not define any statistical model (two-point or otherwise). It is entirely non-parametric. It would be more accurate just to say that MPS does not require the definition of a statistical model.

Line 57: "allow to create" → "allow the creation of"

Line 107: In the methods overview at the end of the introduction, I recommend stating that you will apply a hydrological model to the topography.

Line 125: "exemple" → "example"

Line 139: "This technique allows to co-simulate jointly several variables"

I would elaborate on this sentence for the benefit non-geostatisticians. I think it would be sufficient to say something like "this means that secondary information can be used to improve the simulations."

Line 142: "the use of Gaussian pyramids to account for multiscale patterns"
It would be helpful to provide a brief explanation of what this is and what it accomplishes.

Line 154: "Furthermore, a secondary variable is used during the MPS simulation"

It took me a while to figure out that the secondary variable is the gradient. I would state this more clearly, and explain why it is beneficial to use the gradient.

Line 157: "Two patterns that show the same relative changes even at different absolute altitudes should be considered similar"

Does this mean that the TIs are detrended?

Line 178: "5'000" → "5,000"

There are a few places here where the apostrophe should be replaced by a comma in numbers.

Figure 3. I recommend changing the scale bar label on part A from "variation from the trend" to "difference from the trend" to be more precise.

Figure 4: "Kriegage" → "Kriging." Do the dashed lines represent synthetic GPR surveys? There are three lines in the figure, but in the text it says there are two.

Line 197: "SGS and ordinary kriging are applied using the same variogram model presented in section 2.3"

The variogram in 2.3 is defined for detrended topography, but the synthetic examples are not detrended. How do you justify using the same variogram?

Lines 202-203: "The geostatistical methods that are used to interpolate the basal surface can be used to predict accurately certain derived quantities but not some other quantities"

This is a confusing statement that does not give the reader much information. It might be more helpful to say something along the lines of "We compare the fidelity of the different DEMs by evaluating different performance metrics."

Line 247: "2.5.3 Flow accumulation comparison"

This section needs some motivation for why it is important to accurately represent flow accumulation.

Lines 257-258: "The accumulation is calculated using the Pysheds open source code for watershed delineation."

How does this package compute flow accumulation? Does this package account for ice thickness?

Lines 299-300: "As expected, the kriging estimation produces the smoothest and the SGS the roughest topography."

Why was it expected that SGS would be rougher than MPS? Is there a citation that shows this?

Line 309: "Kriging" → "kriging"

Lines 325-326: "However, the volumes estimated by kriging can over or underestimate the reference, and the method does not provide an error estimation."

See main comments.

Line 340: "Kriging, provides surprisingly a better distribution in these examples"

This is indeed surprising. Do you think this would still be true in areas with sparser bed measurements? It may also be worth discussing the difference in the spatial patterns of the flow paths in Figure 9.

Figure 9: "Krigging" → "Kriging"

Line 385: "Only the SGS and MPS methods are able to estimate the uncertainties on the total volume."

See main comments.

Lines 403-404: "We note that a linear extrapolation of this loss, obviously inaccurate due to all the effects that are not considered in this extrapolation, indicates that the glacier will disappear in about 30 to 40 years."

As the authors have noted, ice loss cannot be accurately linearly extrapolated. As such, I recommend that they remove projections of ice sheet disappearance. Instead, I would emphasize the fact that the glacier has lost a large portion of its volume in a short period of time, and that the proposed interpolation methods could be used to improve estimates of sea level rise contributions from different glaciers.

Line 416: "However, kriging cannot be used to obtain directly the uncertainty on the volume."

See main comments.

Lines 426-427: "Inded, we have shown that MPS provides a much better reproduction of the geomorphology of the simulated basal surfaces"

"Inded" → "Indeed"

It is interesting that SGS does so poorly. Could this be improved by choosing different simulation parameters, such as increasing the search neighborhood? For example, Herzfeld et al., (1993) found that changing the search parameters had a major impact on kriging interpolations.

Herzfeld, U. C., Eriksson, M. G., & Holmlund, P. (1993). On the influence of kriging parameters on the cartographic output—a study in mapping subglacial topography. *Mathematical Geology*, *25*(7), 881-900.

Line 430: "highlithed" → "highlighted"

Line 441: "However, It is" → "However, it is"

Line 447-448: "Finally, when applying existing mass balances to our volume estimation, we were able to draft a possible evolution of the glacier in the context of global warming."

I don't think that the mass loss calculation is enough to say that you can estimate the future evolution. I would instead say that your results indicate that there has been significant mass loss at this glacier, and that these methods enable higher-accuracy ice loss estimates and could enable improvements in glacier retreat projections.

---

## Author Comment (AC1)

**Ice volume and basal topography estimation using geostatistical methods and GPR measurements: Application to the Tsanfleuron and Scex Rouge glacier, Swiss Alps.**

Author(s): Alexis Neven et al.
MS No.: tc-2021-161
MS type: Research article
Iteration: Review

*We thank the two reviewers for their positive evaluation and remarks, which helped us to improve the quality of the paper. Please find below in the left column the reviewers' comments and in the right column the description of how we addressed each comment in the revised manuscript. Spelling mistakes or other small corrections/suggestions regarding the typography style, or word use are not listed in the following description.*

| Reviewer comments | Authors' responses |
|---|---|
| **RC1 on Figure 2:** The label of the time/depth axis is weird, can you explain or modify it? | The figure axis will be modified accordingly. |
| **RC1 Line 410 :** "It might be better if you could quantify such kind of error. I am curious about the influence of velocity error on the ice volume estimation because your TWT is fairly long."

**RC2 :** "1. In my view, the authors should be very careful in interpreting their absolute values of ice volume that they get from all their methods. The primary reason why I am so sceptical is that in L119 the authors say that they used a uniform wave propagation speed (0.168 m/ns) for the time-depth conversion. This is a value that is commonly used for cold ice. However, Tsanfleuron glacier is a polythermal glacier (Hubbard et al. 2003) which means there is a temperate ice layer of significant depth (Schannwell et al. 2014), where this wave propagation speed is certainly lower than what is assumed. While this assumption is absolutely fine for the comparison of the three methods, because the same assumption is used for all of them, it becomes problematic when the absolute ice volume is interpreted. I therefore would like | We agree that we have been too fast on that aspect. Tsanfleuron is a polythermal glacier and using an average constant velocity is an additional source of uncertainty. We estimate that it may change the total volume by about 5%.

As underlined by the RC2, this however does not affect the comparison between the methods since the same assumption is made on all of them.

The reviewers raised an interesting question. To take into account the uncertainty on velocity, it would require to first estimate on the GPR data the thickness of the upper layer, and then spatially invert the depth from a distribution of velocities. Moreover, a better estimation of possible velocities in the particular case of Tsanfleuron (with either Common-Mid-Point GPR measurements or direct thickness measurement) should be carried out.
Finally, comparing the depth derived from such complete analysis and a more classical approach could be really |

| | |
|---|---|
| to see this discussed in more depth and add a few sentences why the presented numbers might be off." | interesting.
We will discuss more carefully this aspect in the revised submission. |
| **RC2 :** Can you comment on how sensitive the MPS is to the selection of your TI?

Given the fact that the TI needs to have similar structures to what is expected under the glacier. How restrictive is this assumption? I guess this is a valid assumption for a small mountain glacier if the lithology does not change? Could this be used for ice sheets? | We agree with the reviewer that the TI impacts the simulations and that this point needs further discussion. We will extend the discussion to cover that aspect.

Briefly, just note that it's possible to compensate partly for a wrong TI by adjusting the parameters. We have shown that in a previous study.

It's also possible to use multiple TIs coming from different possible analogs to account for that uncertainty.

In the case where multiple lithologies are expected under the glacier or the ice sheet, the use of secondary variables, being a lithology identifier, and multiple TIs ((one per type of lithology) could be a solution. We can even give uncertain underlying geology and a probabilistic model for choosing one TI or another. This is already implemented in the code, and will mainly increase the overall uncertainty. |
| **RC2 :** Just to clarify, where ever you have GPR measurements, the interpolated value corresponds to the measured value exactly? | Correct. The Hard Data points are placed in the simulation grid before the MPS simulation. |
| **RC2 :** In Figures 6 and 8, the mean of basal topography for MPS and Kriging look pretty much identical to me. Is this just because for these lines they are not different or do I get almost the Kriging topography if I average over all MPS simulations?

If I do get the same topography, does that mean that the only difference between the two is MPS comes with uncertainty bounds and Kriging doesn't? | It is true that in these two figures, the large-scale cross-sections look similar. At the kilometric scale, they indeed both show the same general trend of the glacier. However be careful, this is not a general result. It is true for this study but it may be wrong if the TI displays different spatial patterns.

So for Tsanfleuron, it is true that Kriging can be sufficient to have a general idea of the regional trend. Kriging allows estimating the uncertainty. However, the uncertainty bounds are usually over-estimated. We propose in this study to look at the uncertainty derived from the SGS approach against the MPS ones.
Even if the large-scale trend looks similar, most of the differences between the |

| | |
|---|---|
| | methods are in the order of a hundred meters distance, or less.
 The best example to illustrate this is Figure 9. Kriging and MPS have a similar trend. At 300m easting, we see an E to W flow, in both Kriging and MPS. However, the connectivity of the cells is very different between the two methods, indicating a very different small-scale topography. |
| **RC2 :** I think the "Conclusions" section needs rewriting. Certainly scratch the first paragraph. In its present form, there seems to be a bunch of different ideas just listed one after the other. My suggestion would be to really highlight the important points:
 • You compare different geostatistical methods
 • Why (and in what situations) is MPS best? What is the drawback of the method?
 • Then highlight where this method could be applied (e.g. boundary condition for glacier models, glacial geomorphology etc.) | Thanks for these suggestions. These points will be included in the revised version of the manuscript. |

| | |
|---|---|
| **RC2 Technical corrections :**
I did not know what conditioning data was at first reading. Is there a more accessible term for this? | We changed it to "field data" in the abstract. |
| L77-80 : This relates to my main concern from above. So based on the presented numbers from 2009 and 2016, the volume has doubled in 7 years. Which of the numbers is more trustworthy? Especially given the fact that your numbers lie pretty much in between these estimates. Why did they have problems with the picking and why did you not have these problems? | The Gremaud and Goldscheider study in 2010 is based on RMT measurements. The apparent resistivity is measured with 4 different frequencies. Then a two-layer inversion is performed, assuming that we have a conductive layer of ice above a resistive layer of limestone. They indicate in their study that it is possible that the inversion has identified the transition between a temperate ice layer, and a more resistive cold ice layer. In addition, in the case of conductive ice, their method (unlike GPR) is poorly sensitive below 100m. Finally, their study is only based on 187 measurement points heterogeneously distributed on the glacier and was interpolated with Kriging. They define 100 Mio $m^3$ as being a minimum value.

Concerning the picking, the GPR wave attenuates as it propagates into the ice, making the deeper part of the bedrock more complicated to map. The identification of a reflector can be affected by the type of equipment used, the amount of water in the ice on the day of acquisition, or the processing for example. We of course also encountered some limitations, and bedrock was not identified in all the lines. |
| L101 : I do not fully follow how you compare absolute volume for a non-glaciated area? I think you say later that you set the surface elevation to 4 m, regardless of the basal topography. Is this correct? Could you make this more clear? | Exactly. We will change the sentence in the revised version of the manuscript. |
| L135 : Does the choice of a random path affect the final result? Would the results be very different if I chose a more regular pattern? | Yes, the choice of the path influences the results, because once a pixel is stimulated, it will influence the next pixels. If they are always simulated in the same order (with a defined path for example), the first pixels to be simulated will always influence the following ones and will lead to a bias or at least a lowered diversity in the simulations. In the literature about MPS there were several studies that tested different types of |

| | paths and the random one is one of the most robust to quantify properly the uncertainty. The results would however not be extremely different, they could have some artifacts due to specific properties of certain paths. |
|---|---|
| L203 : Can you give some examples of quantities that can be predicted and which cannot? | The sentence was not clear and will be rewritten. |
| **RC2** : Figures

Fig. 1 : Why are there white stripes in the aerial image? | We think it is more of an issue with the pdf rendering of the image. We will change the format of the figure. |
| Fig. 3: Hard to tell whether "1/2 (Z(x) ..." is the label of the colourbar in (c) or the label of the y-axis in (d). | It's in fact the label of both. We will correct this in the next version of the manuscript. |
| Fig. 6: To me it looks like kriging has more short-wave variability than the MPS mean. Is there an explanation why MPS is struggling with Test case 19? | It is difficult to say why the MPS is struggling with this particular case. It can be due to a poor representation of the structures present in the test case real topography in the TI.
Concerning the variations, it's actually the other way. Kriging show variations at a higher scale. This is also why it tends to show higher flow accumulation values.
The variations we see in the kriging plot are the influence of new conditioning points entering the range of the variogram and having an influence on the interpolated line. |
| Fig. 7: I'm not sure but shouldn't the SGS value be higher here? | No. SGS simulations are associated with lower cells connectivity (due to more topographic variations). In this context, the probability of having large accumulation values is lower than the other methods. On the other hand, the kriging tends to display smoother results, and therefore has larger cell connectivity, and a higher probability of showing a large flow accumulation value. |
| Fig. 8: Differences in the upper panel are really difficult to see. Maybe you could instead show a difference plot? And why is the kriging topography not shown? | The upper panel was displayed to place the cross-sections and to show that the general trend of the bedrock is similar with both methods. The main comparison support was supposed to be the cross-sections and the flow maps in Fig. 9. We don't think that a point-by-point difference map between methods really reflects the difference in bedrock topography. |

| | Only the average SGS is displayed because it is really close to the Kriging. When the number of simulations is getting large, the SGS means tends to be the best linear estimation, which is the kriging. We will add a note in the revised version of the manuscript. |
| --- | --- |

---

## Author Comment (AC2)

**Ice volume and basal topography estimation using geostatistical methods and GPR measurements: Application to the Tsanfleuron and Scex Rouge glacier, Swiss Alps.**

Author(s): Alexis Neven et al.
MS No.: tc-2021-161
MS type: Research article
Iteration: Review

*We thank the reviewer #3 for her positive evaluation and feedback, which helped us to improve the quality of the paper.*
*Here is a point-by-point answer to the suggestions and comments raised by reviewer #3.*

| Main remarks |
|---|

| | |
|---|---|
| **RC3 :** The authors state that kriging cannot be used to compute ice volume uncertainty, but that is not the case. While it is true that kriging does not sample the uncertainty space in the same way that simulation does, kriging can be used to provide the variance or standard deviation of an estimate at any given location. In theory, multiple realizations produced by sequential Gaussian simulation should converge to a distribution that is represented by the kriging solution. For completeness, I recommend that the authors compute the uncertainty in ice volume for the kriging interpolation. | The reviewer is right when saying that kriging can be used to provide the variance or standard deviation at any given location. However, while we could easily compute the variance map associated with the kriging estimate, this map could not be used to calculate the associated uncertainty on the volume of ice. The volume is the sum of the thicknesses of ice over the map, and therefore the uncertainty is not linearly transposed.

As stated by the reviewer, the mean SGS distribution converges to the kriging estimate. The standard deviations of the SGS are also converging to the standard deviation estimated by kriging. We decided to not represent the variance maps of the kriging estimate in the different figures or cross-sections for more clarity, since they would mainly overlap the SGS ones. Moreover, since these uncertainty maps cannot be used to calculate the ice volume uncertainty, which is one of the main research angles of the study and is also used to calculate scores between the three approaches, we decided to exclude them from the results.

It may be confusing for non-geostatisticians readers and thus we propose to add a few sentences in the review manuscript to |

| | briefly explain this difference and the assumptions behind it. |
|---|---|
| **RC3 :** The authors fit a polynomial trend to the data in order to perform the kriging interpolation and sequential Gaussian simulation. This is a somewhat arbitrary, but often necessary, step for variogram-based methods. How was the degree of the polynomial chosen? Does it matter for your results that the synthetic examples are not detrended? I would like to see more justification for the polynomial selection and discussion on the implications of trend estimation. I also recommend mentioning in the discussion that MPS does not require trend estimation, which is another advantage of the MPS method. | First, we would like to clarify that the synthetic tests also use data that are "detrended". Since the synthetic test data came from the TI, from which the variogram is calculated, we apply the same "detrending" process to these cases in order to be coherent with the rest of the approach. We will clarify that in the manuscript since it is a major point of the approach and that, thanks to the reviewer's comment, appears to be not clearly mentioned.

Regarding the polynomial approach, we do not think that it is relevant to go more in the detail regarding the polynomial trend fitting. The identification of the trend is a common problem in geostatistics. Our approach is to try to find the simplest polynomial degree trend that brings stationarity to the dataset. A simple linear trend was not sufficient, and therefore a cubic one was defined. Choosing a higher polynomial degree may on the contrary over fit the data and remove the variations from the trends which is the variable of interest. |
| **RC3 :** More information is needed on the hydrological modeling method. The Pysheds package that the authors use only has examples for topography without ice. The authors do not state whether or not they account for ice overburden pressure in their hydrological modeling. If that is difficult to do with Pysheds, Chad Greene has a nice tutorial (https://www.mathworks.com/matlabcentral/fileexchange/55352-how-to-estimate-subglacial-wat er-routes). The synthetic examples have a flat ice surface, which could bias the hydrological models. More justification is needed for using the flow accumulation values for the synthetic examples. | The Pysheds package is not used directly for hydrological modeling but only for watershed determination. We only compute the number of cells connected to a cell of interest by following the gradient lines.
The flow accumulation values are indeed calculated on the simulated topographic maps, without the ice coverage and without accounting for the ice pressure.
The idea of this approach is certainly not to estimate a real under-glacial flow model but more to use it as an indicator to compare the structure of the topography and the connectivity of cells.
We totally agree that a calculation of accumulation flow and a calculation of under-glacial flow are not the same, even if there is probably a strong link between the under-glacial topography and the flow.
We will improve the terminology in the revised manuscript, to explicitly point out that we just want to use the connectivity of cells as an indicator for topography comparison. |

| | |
|---|---|
| **RC3 :** I would like to see more discussion on the implications of the hydrological findings. The authors compare the distributions of flow accumulation values for different interpolation methods, but it is unclear why this matters.
Perhaps flow accumulation is important for discriminating between a channelized or distributed drainage system? It may also be helpful to refer to studies by Zuo et al., (2020) and MacKie et al., (2021) which previously investigated the impact of MPS and SGS on hydrological flow. | As stated in the above response, the main scope of the article is to compare the under-glacial modeled topography and the associated multi-level structures. The accumulation of flow is only a highly non-linear indicator, derived from the topography, to compare the realism of interpolation compared to the reference. Of course, we expect that a real under-glacial flow simulation will be influenced by the realism of the simulated topography, but that is not the scope of the paper.
Again, we will make this more clear in the revised manuscript and will refer to relevant studies. |
| **Specific comments** ||
| **RC3 Line 19 :** No comma needed after "crucial". | This will be changed. |
| **RC3 Line 28 :** "Depending of" should be replace by "Depending on". | This will be changed. |
| **RC3 Line 30-32 :** "the choice of the method becomes critical since the flow process is highly non-linear and is strongly linked to the morphology of the subglacial topography" I recommend clarifying that you are referring to hydrological flow, not ice flow (which is also non-linear and dependent on morphology).
What is meant by a non-linear flow process? I think the authors mean that flow accumulation is not a linear function of bed elevation. | We thank the reviewer for pointing out this. The sentence was not clear. We indeed meant that flow accumulation is not a linear function of bed elevation, and this is why we are using this indicator.
We will make it more clear in the revised version of the manuscript. |
| **RC3 Line 35 :** "produces by construction" Awkward wording. It would be sufficient to just say "produces". | This will be changed. |
| **RC3 Line 36-38 :** "Furthermore, even if kriging allows estimation of the local uncertainty on the elevation of the bedrock, it cannot be used to estimate the uncertainty of the global volume of ice (see e.g. Chiles and Delfiner, 2012, p. 478)."

Why can't the uncertainties from kriging be used to estimate ice volume uncertainty? What happens when you use the kriging bed uncertainties to estimate ice volume | It is true that the specific example given by Chiles and Delfiner on page 478 and the following is slightly different from the one of the volume computation but the key idea expressed in this part is applicable to our case. If we want to estimate the uncertainty on the volume, and if we express this uncertainty as a variance, it is clear that the variance of the estimated volume cannot be expressed as a linear function of the topography. Therefore, the kriged map |

| | |
|---|---|
| uncertainty?

The reference to Chiles and Delfiner (2012, p. 478) does not support this statement. Chiles and Delfiner (2012, p. 478) describe a scenario where surface area increases with roughness. It is true that kriging underestimates the surface area of topography (if each grid cell is represented by a tilted plane), but this shouldn't affect the volume calculation. | (even if it provides at any point the variance of the local altitude) cannot be used in a simple manner to get the variance of the volume. Additional information is needed about the covariances between all the pairs of points in the domain. We could write a formal expression of the variance of the estimated volume and it would involve to compute a double integral on all pairs of points in the domain. It is feasible to compute it, using the covariance model or variogram model, but it's not easy and it's for sure not directly computable from the results of the kriging only.

The general message conveyed by Chiles and Delfiner on page 478 and following is therefore applicable here: any non-linear estimation derived from the variable of interest (here the altitude) cannot be simply computed from the kriging map. Either, one has to really pose all the equations and solve them numerically (here we could compute the double integral) or one has to rely on simulations. |
| **RC3 Line 41 :** "Two points spatial statistics" should be "two point spatial statistics".
I recommend elaborating on this sentence so that this concept is more understandable to non-geostatisticians. It might be more understandable to say that these methods are based on variance between pairs of points, and briefly state what a variogram is. | We will include a brief explanation of the principle of a variogram in the revised manuscript. |
| **RC3 Line 51 :** "that the one" should be "that the one". | This will be changed. |
| **RC3 Line 55 :** "require to define" should be changed to "requière the definition of". | This will be changed. |
| **RC3 Line 55 :** "MPS does not require to define an analytical two-point statistics model to represent the spatial variability but infers it in an implicit way".

MPS does not define any statistical model (two-point or otherwise). It is entirely non-parametric. It would be more accurate just to say that MPS does not require the definition of a statistical model | MPS is of course non-parametric but in a sense, the TI is the base of an implicit statistical model. The sentence was written in that sense. |
| **RC3 Line 57 :** "allow to create" should be | This will be changed. |

| | |
|---|---|
| replaced with "allow the creation of". | |
| **RC3 Line 107 :** In the methods overview at the end of the introduction, I recommend stating that you will apply a hydrological model to the topography. | As stated before, we do not apply any hydrological model to the topography but only compute the flow accumulation map in order to compare topographies.

We will make sure that the revised version explicitly states that this accumulation flow is only used as an indicator to compare topographies and not as a real flow model. |
| **RC3 Line 125 :** "exemple" -> "example". | This will be changed. |
| **RC3 Line 139 :** "This technique allows to co-simulate jointly several variables".

I would elaborate on this sentence for the benefit non-geostatisticians. I think it would be sufficient to say something like "this means that secondary information can be used to improve the simulations." | We will modify the text accordingly to the reviewer's suggestion. |
| **RC3 Line 142 :** "the use of Gaussian pyramids to account for multiscale patterns" It would be helpful to provide a brief explanation of what this is and what it accomplishes. | We will add a sentence explaining that Gaussian pyramids simulate patterns at Multi-scale resolution ensuring both large and small patterns have the same quality in their reproduction. |
| **RC3 Line 154 :** "Furthermore, a secondary variable is used during the MPS simulation".
It took me a while to figure out that the secondary variable is the gradient. I would state this more clearly, and explain why it is beneficial to use the gradient. | We will explicitly introduce the secondary variable. |
| **RC3 Line 157 :** "Two patterns that show the same relative changes even at different absolute altitudes should be considered similar".
Does this mean that the TIs are detrended? | The TI is not directly detrended in this approach. However, in order to deal with non-stationary TI, when doing the pattern comparison, we subtract the mean of the pattern. By doing so, we can simulate with non-stationary TI without any trend estimation. |
| **RC3 Line 178 :** "5'000" → "5,000" There are a few places here where the apostrophe should be replaced by a comma in numbers. | The comma is used as the decimal separator in most European countries, and to avoid any confusion we will change it to "5 000" which is the recommended notation. |
| **RC3 Figure 3 :** I recommend changing the scale bar label on part A from "variation from the trend" to "difference from the trend" to be more precise. | This will be changed. |

| | |
|---|---|
| **RC3 Figure 4 :** "Kriegage" → "Kriging." Do the dashed lines represent synthetic GPR surveys? There are three lines in the figure, but in the text it says there are two. | This will be changed. |
| **RC3 Line 197 :** "SGS and ordinary kriging are applied using the same variogram model presented in section 2.3".
The variogram in 2.3 is defined for detrended topography, but the synthetic examples are not detrended. How do you justify using the same variogram? | As explained before the synthetic cases are also "detrended", which justify the application of the same variogram.
We will make sure it is better explained in the revised version. |
| **RC3 Lines 202-203 :** The geostatistical methods that are used to interpolate the basal surface can be used to predict accurately certain derived quantities but not some other quantities".
 This is a confusing statement that does not give the reader much information. It might be more helpful to say something along the lines of "We compare the fidelity of the different DEMs by evaluating different performance metrics. | We agree with the reviewer. The sentence will be changed. |
| **RC3 Line 247 :** The section 2.5.3 "Flow accumulation comparison" needs some motivation for why it is important to accurately represent flow accumulation. | The estimation of under-glacial flow is important for a wide range of applications. We can mention here for exemple prediction of water availability with glacial retreat or modeling of sediment transport. However, the flow estimation itself relies on a good under-glacial topographic estimation. We will make sure that sufficient motivation is added here. |
| **RC3 Lines 257-258 :** "The accumulation is calculated using the Pysheds open source code for watershed delineation."
How does this package compute flow accumulation?
Does this package account for ice thickness? | The accumulation simply computes the gradient of the topography. Then, the connectivity of the cells is computed following this gradient. Local maximums of accumulation are obtained in the topographic minimum since multiple cells are connected to it.
This package does not account for ice thickness. We do not use it to predict an accurate flow, but as a non-linear indicator to compare topographies. |
| **RC3 Lines 299-300 :** "As expected, the kriging estimation produces the smoothest and the SGS the roughest topography."
Why was it expected that SGS would be rougher than MPS?
Is there a citation that shows this? | We propose to add in this part of the paper a reference to Journel and Deutsch (1993), doi: 10.1007/BF00901422.
In this paper, the authors show how the multiGaussian random field model maximizes entropy. |

| | |
|---|---|
| | Because the multiGaussian model maximizes entropy, and because the MPS model honors in general pretty well the histogram and variogram, it means that the MPS model has the same low order statistics as the SGS. But MPS also includes higher-order correlations (less entropy) and is then more likely to produce structures that are globally more structured and connected than the equivalent multiGaussian model generated by SGS. In summary, we expect MPS to perform at least as well as SGS, or better. |
| **RC3 Line 309 :** "Kriging" -> "kriging" | This will be changed. |
| **RC3 Lines 325-326 :** "However, the volumes estimated by kriging can over or underestimate the reference, and the method does not provide an error estimation." See main comments. | See main response to the RF3 general comment. |
| **RC3 Line 340 :** "Kriging, provides surprisingly a better distribution in these examples". This is indeed surprising. Do you think this would still be true in areas with sparser bed measurements? It may also be worth discussing the difference in the spatial patterns of the flow paths in Figure 9. | It is difficult to say but we expect that the kriging will tend to have a bigger difference compared to the reference data in terms of flow distribution if we use sparser conditioning data when the SGS will probably still have a similar distribution. That is probably due to the fact that kriging will tend to be smoother with fewer conditioning points, increasing the connectivity of cells and shifting the distribution up. SGS on the other hand will not produce such effects.

We will complete the section where we comment on the flows. |
| **RC3 Figure 9 :** "Krigging" -> "Kriging" | This will be changed. |
| **RC3 Line 385 :** "Only the SGS and MPS methods are able to estimate the uncertainties on the total volume." See main comments. | See the response to the main comment of the RC3. |
| **RC3 Lines 403-404 :** "We note that a linear extrapolation of this loss, obviously inaccurate due to all the effects that are not considered in this extrapolation, indicates that the glacier will disappear in about 30 to 40 years." As the authors have noted, ice loss cannot be accurately linearly extrapolated. As | We agree that such projection has to be removed and will be in the revised version.

We will emphasize the points suggested by the reviewer, which are also an important aspect. |

| | |
|---|---|
| such, I recommend that they remove projections of ice sheet disappearance. Instead, I would emphasize the fact that the glacier has lost a large portion of its volume in a short period of time, and that the proposed interpolation methods could be used to improve estimates of sea level rise contributions from different glaciers | |
| **RC3 Line 416 :** "However, kriging cannot be used to obtain directly the uncertainty on the volume".
 See main comment. | See the response to the main comment of the RC3. |
| **RC3 Lines 426-427 :** "Inded, we have shown that MPS provides a much better reproduction of the geomorphology of the simulated basal surfaces" "Inded" → "Indeed" It is interesting that SGS does so poorly.
 Could this be improved by choosing different simulation parameters, such as increasing the search neighborhood?
 For example, Herzfeld et al., (1993) found that changing the search parameters had a major impact on kriging interpolations. Herzfeld, U. C., Eriksson, M. G., & Holmlund, P. (1993).
 On the influence of kriging parameters on the cartographic output—a study in mapping subglacial topography. Mathematical Geology, 25(7), 881-900. | The number of nodes does not influence the statistical model.
 Increasing the number of nodes increases the computation time and the simulation quality, but the improvement will reach a plateau. We can denote in table 2 that the difference in terms of indicators between the 12 and the 24 nodes simulations is extremely small, suggesting that increasing the number of nodes will just increase drastically the simulation time without any significant improvements in terms of indicators.
 We also performed some simulation using the Circular Embedding Methods (FFT) that does not depend on node number and takes all the nodes into account. The results were similar.

 Concerning the Kriging tuning, because of the "detrending" process, we are in a stationary process with a mean of zero. We are consequently much less sensitive to parameter influence since the local average estimated during the kriging is always close to zero. In addition, our search distance is almost four times the range of the variogram in order to avoid any sharp artifact when points are entering the search ellipsoid. We think that tuning the parameters will not have any positive influence on the kriging. |
| **RC3 Line 430 :** "highlithed" -> "highlighted" | This will be changed. |
| **RC3 Lines 447-448 :** "Finally, when applying existing mass balances to our volume estimation, we were able to draft a possible evolution of the glacier in the | This will be changed. |

| | |
|---|---|
| context of global warming."
I don't think that the mass loss calculation is enough to say that you can estimate the future evolution. I would instead say that your results indicate that there has been a significant mass loss at this glacier and that these methods enable higher-accuracy ice loss estimates and could enable improvements in glacier retreat projections. | |

On behalf of the authors,

Alexis Neven

---

## Referee Report (RR1)

**Review of "Ice volume and basal topography estimation using geostatistical methods and GPR measurements: Application to the Tsanfleuron and Scex Rouge Glacier, Swiss Alps" revision by Neven et al.**

The revision sufficiently addressed my comments. The hydrological methods description is more complete, and the discussion on ice sheet evolution implications is much stronger. I appreciate the description of how kriging cannot be used to estimate ice volume uncertainty - thank you for teaching me something!

Figure 9d: "Krigging" → "Kriging"

**Line comments**

On a minor note, shouldn't glaciers be plural, even if they are connected?
Title: "Tsanfleuron and Scex Rouge Glacier" → "Tsanfleuron and Scex Rouge Glaciers"
Line 11 and 398: "Scex Rouge and Tsanfleuron Glacier" → "Scex Rouge and Tsanfleuron Glaciers"

Line 115: "estimation of possible flow accumulation" → "estimation of hydrological flow accumulation"

Line 160: "this feature simulate" → "this feature simulates"

Line 177: "this option as the advantage" → "this option has the advantage"

Lines 289-291: "This method is used not to represent precisely the possible flow at the base of the glacier but to compare easily and rapidly the results of the application of …"

The wording is a bit challenging. I recommend rewording it to something like this: "This method is not used to precisely represent hydrological flow at the base of the glacier. Rather, it is used to easily and rapidly compare the results of the application of…"

Line 292: "cells connectivity" → "cells' connectivity"

Line 419: "5.2 Ice volume of the Tsanfleuron Glacier"
Shouldn't this include Scex Rouge in the heading?

Line 497: "Kriging and SGS require to analyze…" → "Kriging and SGS require the analysis of…"

Line 513: "Training Image"
This is the only place where training image is capitalized. I recommend making it lower case for consistency.

---

## Author Response (AR2)

**Ice volume and basal topography estimation using geostatistical methods and GPR measurements: Application to the Tsanfleuron and Scex Rouge Glaciers, Swiss Alps**

Author(s): Alexis Neven et al.

MS No.: tc-2021-161
MS type: Research article
Iteration: Final Review

*We thank the reviewers and the editor for their feedback on our revisions.*
*We implemented all the corrections suggested by the reviewers and the editor and performed a last careful read. Some additional minor typos were also corrected.*

*On behalf of the authors,*
*Best Regards*

*Alexis Neven*